



# Sources of Uncertainty in Greenland Surface Mass Balance in the 21[st] century.

Katharina M. Holube[1,3], Tobias Zolles[1,2], and Andreas Born[1,2]

[1]Department of Earth Science, University of Bergen, Bergen, Norway
[2]Bjerknes Centre for Climate Research, University of Bergen, Bergen, Norway
[3]now at: Meteorological Institute, Universität Hamburg, Hamburg, Germany

**Correspondence:** Andreas Born (andreas.born@uib.no)

**Abstract.** The surface mass balance (SMB) of the Greenland Ice Sheet is subject to considerable uncertainties that complicate predictions of sea level rise caused by climate change. We examine the SMB of the Greenland Ice Sheet in the 21[st] century with the surface energy and mass balance model BESSI. To estimate the uncertainty of the SMB, we conduct simulations for four greenhouse gas emission scenarios using the output of a wide range of climate models from the sixth phase of the Coupled

Model Intercomparison Project (CMIP6) to force BESSI. In addition, the uncertainty of the SMB simulation is estimated by using 16 different parameter sets in our SMB model. The median SMB across climate models and parameter sets, integrated over the ice sheet, decreases over time for every emission scenario. As expected, the decrease in SMB is stronger for higher greenhouse gas emissions. The regional distribution of the resulting SMB shows the most substantial SMB decrease in western Greenland for all climate models, whereas the differences between the climate models are most pronounced in the north and

in the area around the equilibrium line. Temperature and precipitation are the input variables of the snow model that have the largest influence on the SMB and the largest differences between climate models. In our ensemble, the range of uncertainty in the SMB is greater than in other studies that used fewer climate models as forcing. An analysis of the different sources of uncertainty shows that the uncertainty caused by the different climate models for a given scenario is larger than the uncertainty caused by the climate scenarios. In comparison, the uncertainty caused by the snow model parameters is negligible, leaving the

uncertainty of the climate models as the main reason for SMB uncertainty.

## 1 Introduction

The Greenland ice sheet (GrIS) experiences a net mass loss through changes in surface mass balance (SMB) and dynamical processes such as solid ice discharge: In 2005-2017, the GrIS contributed almost as much to sea level rise as all glaciers world-wide (Sasgen et al., 2020). There is substantial uncertainty in the magnitude of sea level rise that will be caused by the GrIS

in the future (Goelzer et al., 2020). According to Slater et al. (2020), the contribution of melt to sea level rise in 2007-2017 exceeded the highest estimates of the IPCC Fifth Assessment Report sea level predictions, whereas for dynamic ice loss the lower or middle estimates are met. The influence of SMB on the total mass loss becomes more important in the future because outlet glaciers will retreat above sea level (Fettweis et al., 2013). The uncertainty in ice discharge is not as substantial as the





uncertainties of climate projections and in SMB (Aschwanden et al., 2019).

SMB simulations are subject to uncertainty from multiple sources, such as the spatial resolution of the ice sheet model, the parametrisation of processes like melt-albedo feedback and the forcing of the SMB model (Goelzer et al., 2013). To assess the uncertainty of the forcing, we consider four future pathways with different radiative forcings (hereafter climate scenarios) that lead to different extents of climate change. We simulate the SMB using the output of a wide range of climate models from
the sixth generation of the Coupled Model Intercomparison Project (CMIP6) to take into account the uncertainty of climate projections for a specific climate scenario, although the similarities of the climate models limit the validity of this approach (Knutti et al., 2013). To estimate the uncertainty of the parametrisation, we conduct all simulations with several sets of parameters of the SMB model BESSI (Born et al., 2019; Zolles and Born, 2019). While this approach cannot substitute a comparison of different SMB models as in Fettweis et al. (2020), it enables us to assess the relative importance of climate and snow-related
parameters in a coherent framework.

The SMB of the GrIS has been simulated with models of different complexities. PDD models apply an empirical relationship between melt and temperature. Several climate models from CMIP3 have been used to force an ice sheet model in which the SMB is calculated by the PDD method (Graversen et al., 2011). Yan et al. (2014) employed another ice sheet model that also
uses the PDD method for the SMB calculations and forced it with CMIP5 climate models. However, PDD models are calibrated to match the present state of the climate and so their validity in a warming climate is limited (Vizcaino, 2014). This is less of a concern in regional climate models (RCMs), coupled with a snow model where many physical processes are resolved. However, these models are computationally expensive, leaving their use to evaluating only a few climate models (Fettweis et al., 2008; Franco et al., 2011; Fettweis et al., 2013; Hanna et al., 2020). In Fettweis et al. (2008), a multiple regression for
the SMB changes as a function of temperature and precipitation is performed to calculate the SMB changes also for the CMIP3 models that they have not simulated. For CMIP6, Hanna et al. (2020) simulated the SMB of Greenland using the output of five climate models. Hofer et al. (2020) showed that the predicted climate from these representatively selected climate models leads to a larger GrIS SMB decrease in CMIP6 than in CMIP5. While their results already include some variability between climate models, their selection from the CMIP6 model pool is necessarily incomplete, and the relative importance of climate
simulation as compared with other sources of uncertainty remains unclear.

We address some of those open questions in this study with the surface energy and mass balance model "BErgen Snow SImulator" (BESSI) (Born et al., 2019). We simulate the SMB of Greenland for 26 climate models from CMIP6, four climate scenarios and 16 parameter combinations of BESSI to quantify and compare the different uncertainties. In addition, we study
spatial variations in the simulated SMB and the importance of the different input variables in different parts of Greenland (Sect. 3), after a description of our methods (Sect. 2). Finally, we compare our results to previous studies (Sect. 4).





## 2 Methods

### 2.1 Snow Model

The BErgen Snow SImulator (BESSI) (Born et al., 2019) is a surface energy and mass balance model for glaciated regions
with a flexible spatial domain. In this study, the domain is Greenland with an equidistant resolution of $10\,\mathrm{km}$. The topography
of the ice sheet is based on ETOPO (Amante and Eakins, 2009) and remains fixed throughout the simulations. The vertical
dimension consists of up to 15 layers that are adjusted depending on the snow in each grid cell. The five daily input variables
are air temperature and dew point at $2\,\mathrm{m}$ above ground, the amount of precipitation, and surface downwelling shortwave and
longwave radiation. The top layer changes its mass and energy according to the the forcing of the input variables. Precipitation
falls as snow when the air temperature is below $0\,^\circ\mathrm{C}$, and as rain otherwise. Melt water percolates down into deeper layers and
refreezes. Horizontal exchanges of mass or energy are deemed negligible on the $10\,\mathrm{km}$ grid. When there is no more snow left to
melt, the excess energy is used to melt ice. Corrections are made when the melt exceeds the existing amount of ice (Appendix
A). For a detailed description of the snow model, see Born et al. (2019) and Zolles and Born (2019). The performance of BESSI
has been compared with other SMB models in Fettweis et al. (2020).

BESSI uses parametrisations of e.g. snow albedo or the turbulent heat exchange. The model parameters of BESSI are
calibrated to RACMO (Noël et al., 2016) focusing on a single optimal solution. Here, we use an ensemble of equally plausible
model parameter settings based on a multivariate calibration (Zolles et al., 2019). For the calibration, BESSI was run for 500
years with ERAinterim as forcing data using different parameter combinations. The performance of the simulation is compared
to RACMO over the period 1979-2017 on an annual basis. We are using seven measures of goodness of fit, based on the bias,
the mean absolute deviation (MAD) and the root mean square error (RMSE) of the SMB. The bias is the difference between the
ice-sheet wide integrated mass balance between RACMO and BESSI, while we calculate three representations of RMSE and
MAD. The first calculates the Greenland wide SMB and its temporal MAD over the years, the second one calculates a temporal
MAD for each point and averages them over all grid points, with the last being the MAD over all points in space and time. A
similar approach is used for the RMSE. This gives us seven objective functions for the multivariate optimization. Similar to
the method used by Zolles et al. (2019) we calculate the Pareto optimal set. This yields a total of 16 different solutions with a
fresh snow albedo between 0.766 and 0.891, a firn albedo between 0.480 and 0.696, and a turbulent heat exchange coefficient
between 5.2 and $12.2\,\mathrm{W\,m^{-2}\,K^{-1}}$ (Zolles and Born, 2019). All optimal solutions use the same albedo routine from Bougamont
et al. (2005).

### 2.2 Climate Models

We use climate model output of CMIP6 for the period of 2015-2100 (Eyring et al., 2016). The Tier 1 scenarios (with increasing
radiative forcing: SSP126, SSP245, SSP370 and SSP585) from ScenarioMIP are selected for this study because they encom-
pass a wide range of future forcing possibilities (O'Neill et al., 2016) and are available for many different climate models. We
selected 26 climate models that provide all of BESSI's input variables for at least two scenarios (Table Appendix B1), whereas

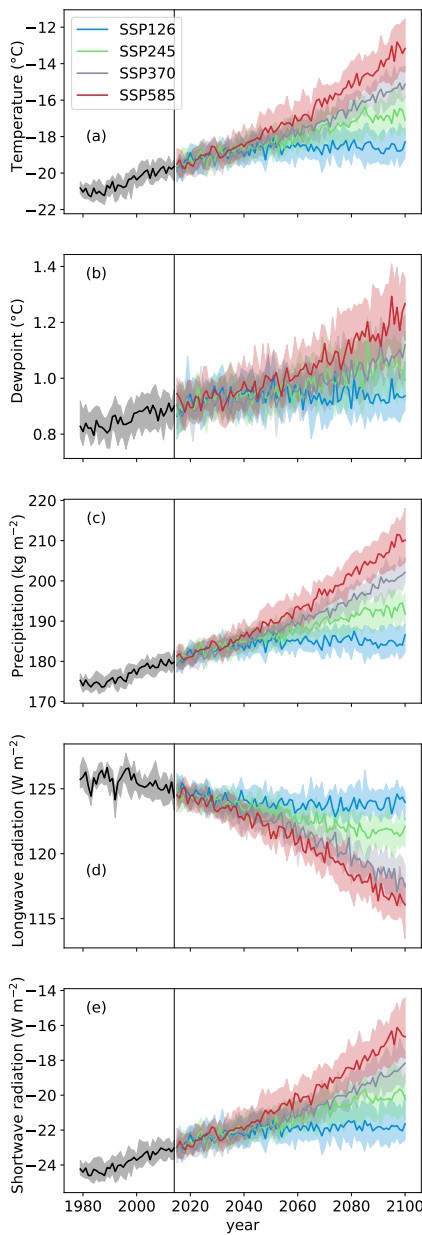

**Figure 1.** Input variables for BESSI for different scenarios, averaged over the Greenland ice sheet. The solid line is the median over all climate models for one scenario, the shaded area between the 25 % and 75 % percentiles represents half of the climate models. (a) Temperature in 2 m above ground. (b) Dewpoint in 2 m above ground. (c) Amount of precipitation. (d) Surface downwelling longwave radiation. (e) Surface downwelling shortwave radiation.





the dew point is calculated from the relative humidity if necessary.

The input variables are interpolated linearly to the 10 km BESSI grid. Climate model biases are calculated based on the delta method (Beyer et al., 2020) by comparing the daily mean of the historical simulation and the mean of the ERAinterim reanalysis data in the period of 1979-2014 (Dee et al., 2011). For all input variables except precipitation, the differences be-

tween the daily means are subtracted from the future projection. Precipitation is bias corrected by the ratio of ERAinterim and historical mean precipitation because its high variability would lead to negative values if the difference was used. During the winter, shortwave radiation may be very weak so that the bias correction can lead to localised, small negative values. These values are set to zero. The daily means of precipitation are affected by individual intense precipitation events due to the short length of the historical period. The monthly biases are less affected by individual high precipitation events and therefore we

multiply the projected precipitation with the ratio of the monthly means instead of the daily means.

Throughout the 21$^{\text{st}}$ century, the median air temperature over all climate models rises in every scenario except in the scenario with the smallest increase in greenhouse gases (SSP126), where it remains almost constant during the second half of the century (Fig. 1a). While shortwave radiation decreases slightly, precipitation, longwave radiation and dew point increase over the course

of the century (Fig. 1b-e). The stronger the greenhouse gas forcing, the larger the increase in these variables. Precipitation is the variable where the range of values caused by the climate model differences overlap most for the different scenarios (Fig. 1c).

### 2.3   Simulations and Ensemble Design

We conduct two different kinds of SMB simulations: The main ensemble consists of simulations for different climate models,

scenarios and snow model parameter sets. It illustrates the temporal and spatial behaviour of the SMB and it enables us to separate the different uncertainty components. The "single forcing" ensemble shows the influence of the individual input variables.

The main ensemble uses 96 selected climate model-scenario combinations. In addition, we conduct 26 simulations for the historical reference period (1979-2014), i.e. one for each climate model. Each of the simulations is conducted with 16 different

snow model parameter sets, resulting in 1952 simulations. The selection process of the parameter combinations is described in Sect. 2.1. The firn cover is initialised by forcing BESSI with ERAinterim reanalysis data for 540 years, to reach a dynamically and thermodynamically stable firn cover at the year 2014. The long response time of the firn cover requires an initialisation period of several hundred years, which is realised by forcing the model with the ERAinterim data 15 times back and forth (Zolles and Born, 2019). For the historical time period, the initialization ends in 1979 after 14 ERAinterim cycles back and

forth. For every parameter set, the same initialised firn cover is used to save computation time.

In the single forcing simulations, the transient climate model simulations are used as input for only one variable, and the daily ERAinterim climatology for the others to assess the influence of each variable on the SMB. The scenario SSP585 is chosen





because it is available for all 26 climate models, and we used the snow model parameter set that produces the best results in the

calibration with RACMO (Sect. 2.1). For precipitation, the daily ERAinterim climatology cannot be used as it overestimates

the surface albedo due to small amounts of snowfall every day. This leads to an overestimation of the mass balance of up to

40 % (Zolles and Born, in prep). Instead we use the monthly precipitation climatology and distribute the ERAinterim monthly

average $P_{\mathrm{ERAi}}^m$ following the distribution of the actual climate simulation:

$$P_{\mathrm{year,\,clim}}^d = \frac{P_{\mathrm{year}}^d \cdot P_{\mathrm{ERAi}}^m}{P_{\mathrm{year,\,model}}^m} \tag{1}$$

where $P$ stands for precipitation, $m$ for monthly mean, $d$ for daily mean and year stands for the point in time of the simulation.

Thus, the climatological daily precipitation distribution differs for each climate model, but the monthly averages are the same.

For each of the 26 climate models, we conducted 6 simulations for the SMB: a reference simulation with the historical cli-

matology and 5 simulations with different transient variables. We need a separate reference simulation for each climate model

because the precipitation distribution differs for each climate model according to Eq. 1.

## 3 Results

### 3.1 Scenario Surface Mass Balance Simulations

In this section, we show temporal and spatial differences between the climate models and climate scenarios of the median

SMB over all parameter combinations. The median SMB at the end of the century over the climate models and snow model

parameters is shown for the different climate scenarios in Table 1. The surface mass balance decreases relative to the historical

simulations in all scenarios (Fig. 2). In the moderate scenario SSP126, the SMB is relatively stable to the end of the century.

Higher emissions of greenhouse gases (stronger forcing) lead to a lower SMB (SSP245, SSP370, SSP585). With greater

warming, the range in simulated SMB for different climate models increases, although the range in input variables except

precipitation is of the same magnitude for all scenarios (Fig. 1). For precipitation, the interquantile range between the climate

models increases only slightly with stronger greenhouse gas forcing. Precipitation variability alone cannot explain the larger

interquantile range in SMB in the warmer scenarios. The reason for the increasingly dissimilar SMBs with stronger greenhouse

gas forcing is the larger temporal change in the input variables, because a combination of several distinct changes in the input

can intensify the SMB change (Sect. 3.3). When the snow model is forced with ERAinterim data (Fig. 2, orange), a relatively

low SMB in the early 21$^{\text{st}}$ century is apparent. This correlates with more frequent Greenland blocking (Sasgen et al., 2020).

A similar reduction in SMB is not observed when forcing BESSI with historical climate model data (Fig. 2, black), probably

because the climate models do not reproduce the observed increase in blocking activity (Davini and D'Andrea, 2020).

Spatial anomalies for the last decade of the SMB in the low emission scenario SSP126 and the high emission scenario

SSP585 are shown in Fig. 3. In the west of Greenland, the SMB in the 2090s is lower than in ERAinterim (1979-2014),

independent of the scenario (Fig. 3a, b). In this region, higher temperatures lead to increased melt. In the centre of the ice





**Table 1.** Median and quartiles over all climate model and snow model parameter combinations of the 2091-2100 SMB mean value for different scenarios.

| Scenario | Historical | SSP126 | SSP245 | SSP370 | SSP585 |
|---|---|---|---|---|---|
| Median SMB (1979-2014 or 2091-2100) / $\mathrm{kgm}^{-2}$ | 399 | 318 | 254 | 42 | -226 |
| 75 % quantile SMB / $\mathrm{kgm}^{-2}$ | 415 | 384 | 308 | 194 | -1 |
| 25 % quantile SMB / $\mathrm{kgm}^{-2}$ | 378 | 257 | 104 | -308 | -623 |

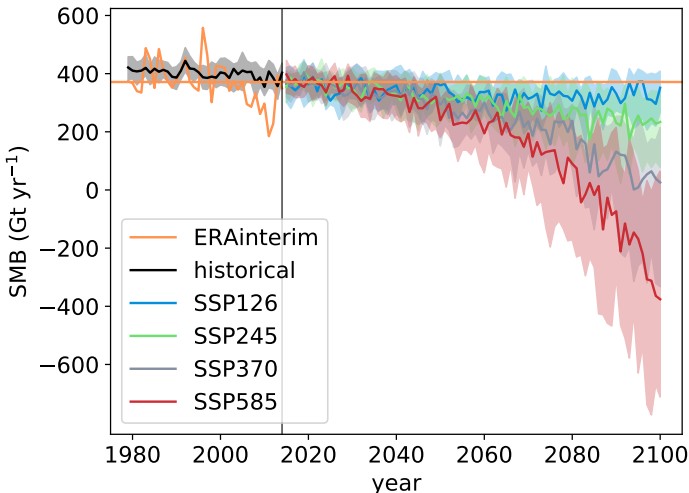

**Figure 2.** Surface mass balance simulations forced with ERAinterim reanalysis data, historical climate simulations and scenario climate simulations, median over the SMB for all snow model parameter combinations. The solid line is the median of all climate models, the shading the 25 % and 75 % percentiles. Orange: SMB forced with ERAinterim with mean value.

sheet, the SMB is slightly higher than in ERAinterim, especially in the southeast. There, an increase in precipitation with a warming climate is dominant. However, the SMB increase in the centre is greatly outweighted by the SMB decrease at the margin of the ice sheet. Both SMB changes are much more pronounced in the high emission scenario SSP585 because of the greater increase of the input variables. (Shortwave radiation decreases over time but it has almost no influence on the SMB, Sect. 3.3.) Already today, SMB changes are dominated by melt in the west and by snowfall in the east (Sasgen et al., 2020). In

the north, the temperatures are too low for much melt at the present day, but with an average increase of temperature over the ice sheet of approximately 6 K in SSP585 (Fig. 1a), melt increases considerably. At the margin of the ice sheet, the standard deviation of the SMB between the climate models is largest (Fig. 3c, d). Because of the high melt rates at the margin, the relative standard deviation is greatest near the equilibrium line (Fig. 3e, f), which means that the SMB is most sensitive to the choice of climate model in that region. In the high emission scenario SSP585, the equilibrium line varies substantially more


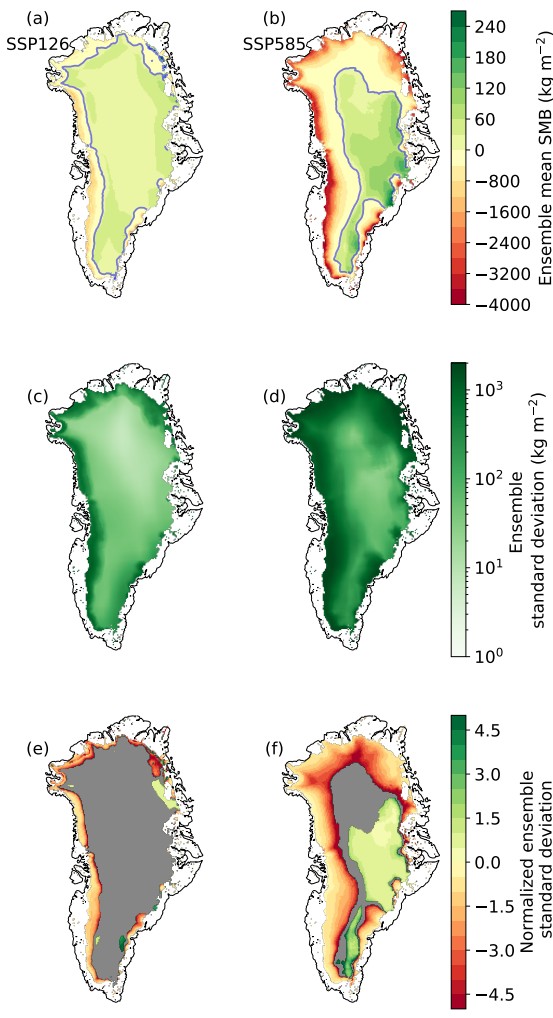

**Figure 3.** Anomaly of the median SMB over all parameter combinations (2090-2099 mean) with respect to ERAinterim (1979-2014 mean) (a, b) with standard deviation (c, d) and relative standard deviation (e, f) for the scenarios SSP126 (a, c, e) and SSP585 (b, d, f). (a, b): The contour line indicates a mass balance of zero. Note the different scales for positive and negative values. (e, f): In the grey area the absolute value of the surface mass balance is smaller than 50 $kg/m^2$, and for values near zero no meaningful relative standard deviation can be calculated.

between climate models than in the moderate scenario SSP126 (Fig. 4). Equilibrium line changes show that the differences between climate models increase with stronger greenhouse gas forcing (Fig. 2).



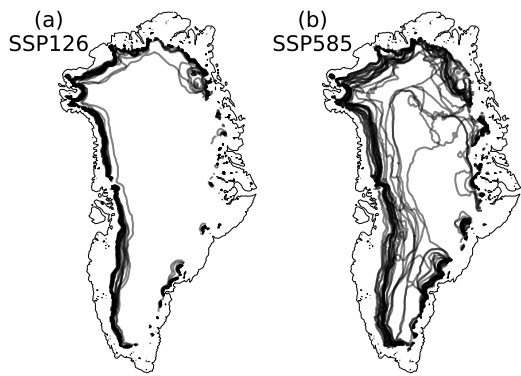

**Figure 4.** Equilibrium lines of the median SMB over all parameter combinations (temporal mean for the period of 2090-2099) for different climate models and the scenarios SSP126 (a) and SSP585 (b).

## 3.2 Estimation of Uncertainties

Having examined the spatial variations between climate models, we next study the variance of the full ensemble containing climate models, emission scenarios and snow model parameters. We split the total uncertainty of the simulations into four
different components: climate model, climate scenario and snow model parameter uncertainty and internal variability. The method used is based on Hawkins and Sutton (2009) and described in Appendix C: A forth degree polynomial fit is applied to the decadal running mean of the SMB to separate the climate model, scenario and parameter dependencies from the residuals of the fit, which are considered as the internal variability of the system. The law of total variance is applied to the whole ensemble of the polynomials to separate the variances of the three components for each year. These variances quantify three relevant
sources of uncertainty, with internal variability being the forth.

The sum of the different uncertainty components increases strongly over the course of the century (Fig. 5a). The relative contributions of the different uncertainty components are better visible when normalised with the sum of all components (Fig. 5b). In the first years of the simulations, the internal variability is the largest source of uncertainty, showing that it is most
important in the absence of external forcing. While the scenario uncertainty has the smallest contribution in the beginning, its importance increases in the second half of the century, as decarbonisation measures and the adaption of the climate system take time (Davy and Outten, 2020). The parameter uncertainty is slightly larger than the scenario uncertainty at first, but its relative importance decreases in time. Its overall small contribution to uncertainty indicates that the results of our SMB simulations are almost independent of the specific parameter combination of BESSI. The parameter uncertainty does not depict the total snow
model uncertainty, because the approach to calculate the SMB is the same regardless of the parameter combination, whereas differences in the climate models are caused by different ways of simulating the processes. A few years into the simulation, the climate model uncertainty becomes the largest contributor to the uncertainty and the share of the internal variability decreases



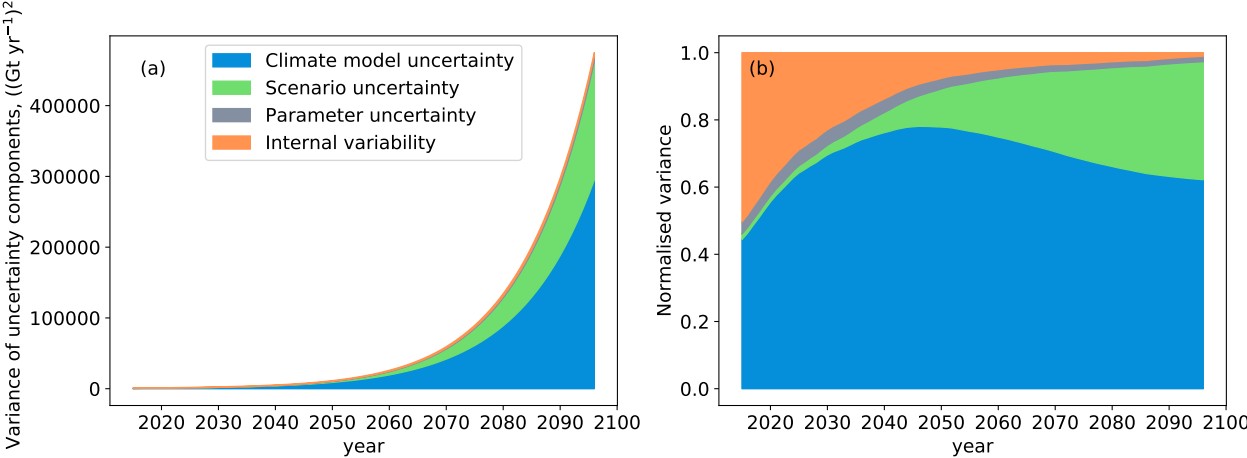

**Figure 5.** (a) Total and (b) relative variances of the different uncertainty components: Choice of climate model (blue), different emission scenarios (green), different snow model parameters (grey), and internal variability (orange).

rapidly. At the end of the century, the climate model uncertainty is still greater than the scenario uncertainty. In the scenarios with strong forcing, there are some climate models that induce only small SMB changes, while other climate models lead to a
much stronger SMB decrease. This great variability is larger than the differences of the model consensus between the scenarios. At the end of the century, the climate model uncertainty is about 62 % and the scenario uncertainty is about 35 % of the total variance, whereas the snow model parameter uncertainty and the internal variability represent about 3 % combined.

The separation of variances can be generalised to every grid cell of the GrIS. The total variance of the 1952 simulations is
largest at the margin of the ice sheet, where the SMB changes most (Fig. 6a, b). The total variance increases by several orders of magnitude from the middle to the end of the century. At the middle of the century, the climate model uncertainty is the most important component at the margin and in the centre of the ice sheet (Fig. 6c). Only in the north and at higher altitudes in the west, the internal variability is largest. Compared to the other components, the scenario uncertainty is insignificant at the middle of the century (Fig. 6e). At the end of the century, the scenario uncertainty becomes more pronounced, especially at
the western margin, where the amount of melt differs considerably between the scenarios (Fig. 6d). The area where the climate model uncertainty has the largest share increases even more at the end of the century, mainly at the expense of the regions where the internal variability is important at the middle of the century. Only at the margin of the ice sheet and in the area where the total variance is low, the scenario uncertainty has a similar magnitude as the climate model uncertainty.

### 3.3   Single Forcing and Regional Analysis

In the single forcing simulations, we run the snow model using only one input variable from each CMIP model simulation. This variable is hereafter called the transient variable. For the other variables, the ERAinterim mean of the historical period is used





in the simulation, except for precipitation, whose temporal distribution is again adapted as described in Sect. 2.3. We study the influence of the different input variables on the SMB for the total GrIS and three of the regions which were previously used in Zolles and Born (2019) (Fig. 7). These regions are selected because they illustrate the spatial differences in the behaviour of
the SMB.

The SMB increases when precipitation is the transient variable due to an increase in snowfall (Fig. 1c). In the simulation with transient dew point, the SMB also increases through an increase in resublimation, but the effect is smaller. When the downwelling longwave radiation increases, the snow temperature rises, which leads to more melt. The effect of melt caused
by increased air temperature is stronger than that of increased longwave radiation except for the east where the SMB change is dominated by precipitation changes. The interquantile range is largest when temperature or precipitation are the transient variables except for the east where the dew point has a larger interquantile range than the temperature. Consequently, these variables dominate the uncertainty of the SMB simulations. The sum of all individual changes does not equal the fully transient simulation (Fig. 7) which is the same as the SSP585 scenario of the first ensemble (Fig. 2), pointing toward important non-
linearities that amplify the SMB reduction: For example, air temperature and precipitation often covary so that the increased precipitation compensates the increased melt only to a certain degree, because additional heat is added to the snow when more precipitation falls as rain in a warming climate. When air temperature and longwave radiation rise together in a warmer and cloudier future, more energy is available at the surface and due to the non-linearity of the SMB, increased melt is observed than from each of these forcing components individually. The impact of the increasing amount of longwave radiation decreases
with rising surface temperature because the net flow of sensible heat depends on the vertical temperature gradient. The amount of resublimation is larger if only the dew point is increased, while it gets less if the surface gets warmer. The combination of these non-linear effects lead to a larger interquantile range in the fully transient simulations compared to the ranges in the simulations with the transient input variables.

In the western region, the SMB and its different components follow a similar course as for the entire GrIS, except for the amount of SMB decrease per area which is in the fully transient simulation about five times as high (Fig. 7). Additionally, the internal variability is not as important as in the total GrIS, and the scenario uncertainty is slightly higher (Fig. 8a). This shows a high dependence of surface melt on the climate scenario in this region.

In the northern region, the SMB increases with transient precipitation and transient dew point to the same extent (Fig. 7c). Resublimation and sublimation are important contributors to the SMB in this dry region. This is in line with Box and Steffen (2001) who show that 28 % of the accumulation is caused by resublimation at one station in the northeast at 2113 m above sea level. Even the precipitation increase will not dominate in the north by the end of the century. In the fully transient simulation and in the simulation with transient temperature, the SMB decreases strongly and non-linearly at the end of the century (Fig.
7c, orange line). The decrease in SMB is rather late because of the low temperatures in the north at present day. However, when the temperatures rise high enough, they trigger a strong albedo feedback because snowfall is scarce. The uncertainty associated





with the choice of climate model has a larger share in the north than Greenland wide because the temperature differences between climate models are more pronounced, which suggests discrepancies in the simulated sea ice cover. As a consequence, the scenario uncertainty has a smaller impact (Fig. 8b).


In the east, the SMB with transient precipitation follows the SMB with all variables transient closely, showing that the main cause for SMB changes is the precipitation (Fig. 7d). Fettweis et al. (2013) also found increased precipitation in the east because the reduced sea ice cover leads to a moister atmosphere. The uncertainty ranges between climate models for transient precipitation and the fully transient simulation are also very similar, therefore the climate model uncertainty is mostly

a precipitation uncertainty. The internal variability has a large contribution to uncertainty (Fig. 8c) because the total uncertainty of all other components is small (not shown). The climate model uncertainty is still the largest component, showing an increase at the end of the century (Fig. 8c) when the fully transient SMB stagnates (Fig. 7d).

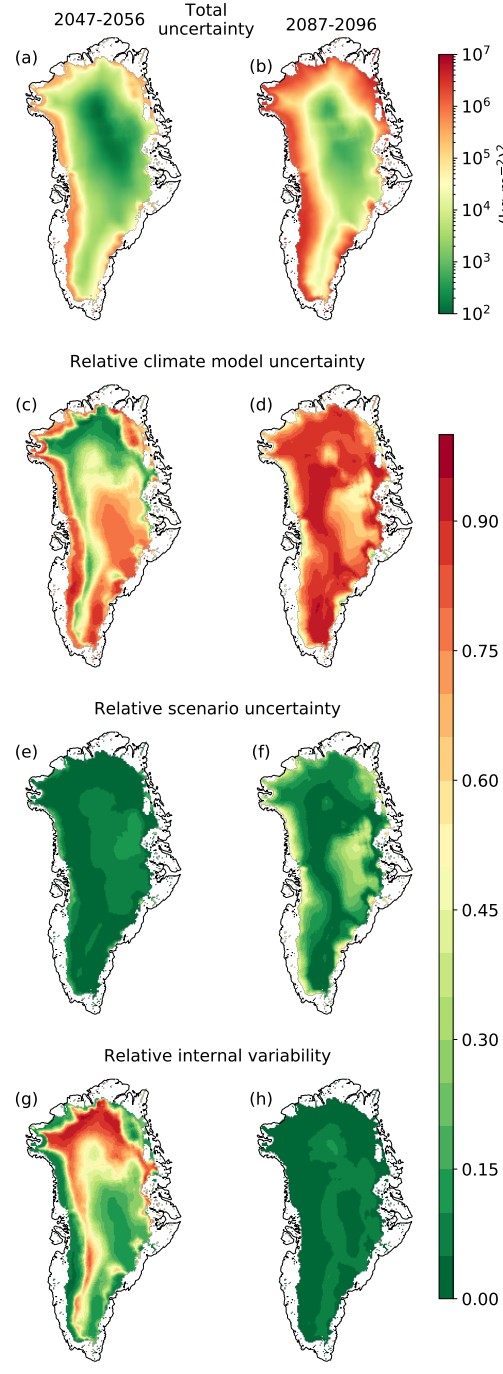

**Figure 6.** (a, b) Total variance, consisting of climate model uncertainty, scenario uncertainty, snow model parameter uncertainty and internal variability. (c, d) Ratio of climate model uncertainty and sum of the uncertainties. (e, f) Ratio of scenario uncertainty and sum of the uncertainties. (g, h) Ratio of internal variability and sum of the uncertainties. (a, c, e, g) show the mean over the years 2047-2056, (b, d, f, h) the mean over the years 2087-2096. The latter are the last 10 years of the analysis because the decadal mean SMB is not defined for 2097-2100.

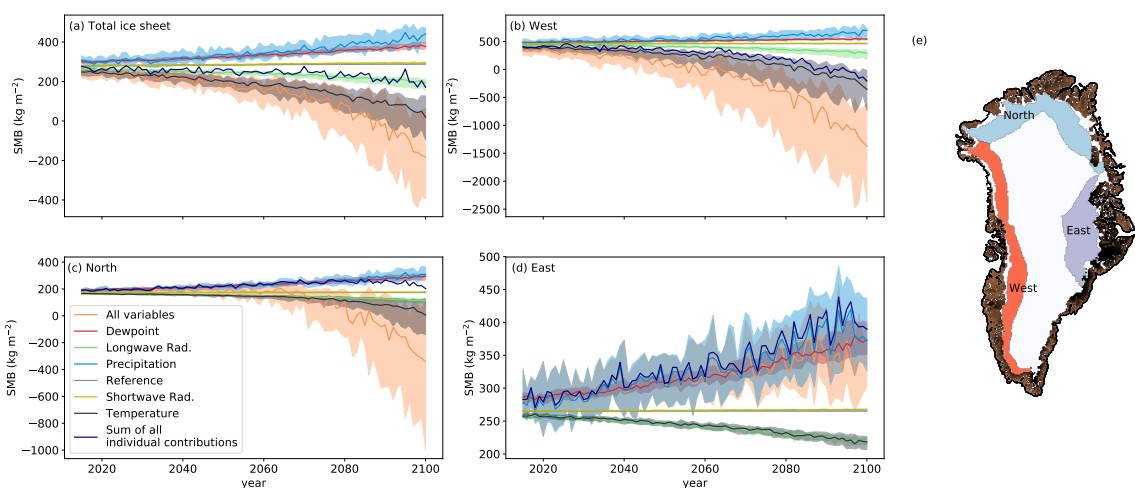

**Figure 7.** SMB for single forcing simulations, for the entire GrIS (a) and selected regions (b-d). The variable named in the legend is transient for scenario SSP585, while all other variables are the ERAinterim mean. "All variables": all variables transient, same as Fig. 2. "Reference": historical climatology for all variables with precipitation distribution as in CMIP. (e) Positions of the selected regions. Regions "North" and "West" are at elevations of 1000-2000 m. The southeast is precipitation driven and the change in SMB with altitude is less developed, therefore the region "East" is at elevations of 1000-3000 m.


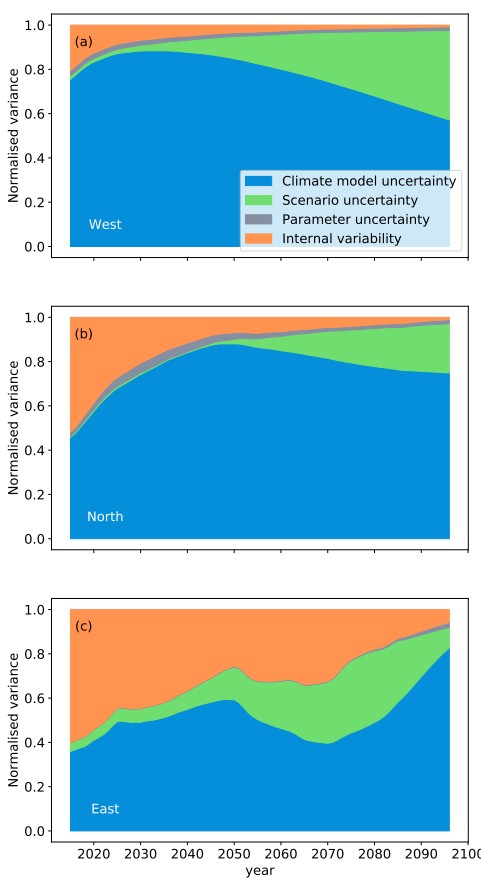

**Figure 8.** Relative variances of the different uncertainty components for 3 different regions of the GrIS: Uncertainty associated with the choice of climate model (blue), uncertainty caused by different emission scenarios (green), uncertainty of different parameter combinations of the snow model (grey), and internal variability (orange), being the variance of the residues of a forth degree polynomial fit to the decadal mean integrated SMB. The calculations are described in Appendix C.





## 4 Summary and Discussion

We simulated the SMB of the GrIS with the snow model BESSI for most of the available climate simulations in the CMIP6

database, using four different climate scenarios, and 16 parameter configurations of our snow model. In the high emission scenario (SSP585), the surface mass loss accelerates and the integrated SMB is about -230 $\mathrm{kg/m^2}$ at the end of the 21st century, whereas in the low emission scenario SSP126 the integrated SMB is only slightly lower than in the historical time period and approximately constant (Table 1, Fig. 2). Taking into account the ice discharge, which amounts to near 500 $\mathrm{Gtyr^{-1}}$ between 2005 and 2019 (Mankoff et al., 2020), our historical simulations result in a negative total mass balance. Assuming

an approximately unchanged discharge, the lower SMB in the scenario SSP126 would lead to substantial mass loss in the future.

The regions with the most pronounced changes in SMB are the west and the north of Greenland. In the west, the SMB is already dominated by melt, and in the north, additional melt is not fully compensated by the scarce precipitation. In the east, we simulate a higher SMB than at present day because of a warmer and moister climate in future projections. We find that the

choice of climate model has the largest overall influence on the uncertainty in SMB projection, exceeding even the variance between climate scenarios. This effect is localised mostly near the the equilibrium line.

The results presented here are in good agreement with previous studies. All ice sheet models in Goelzer et al. (2020) simulated an accelerated mass loss with stronger greenhouse forcing. They used the high-end scenario in CMIP5 with a repre-

sentative concentration pathway (RCP) that leads to a radiative forcing of 8.5 $\mathrm{Wm^2}$ at the end of the 21st century (RCP8.5), comparable to the SSP585 pathway we used here. Detailed SMB estimates are also available from the regional climate model MAR forced by a selection of CMIP6 global climate models (Hanna et al., 2020). This study also finds the familiar acceleration in mass loss. However, four of the five climate models used to force MAR have an above-average equilibrium climate sensitivity (ECS, Meehl et al. (2020)), so that changes in temperature are exacerbated. Comparing our simulations with those

of MAR that were forced by the same CMIP6 models, we find that in four out of five cases BESSI simulates a higher SMB than MAR (Fig. 9a). This is plausible because BESSI has a stronger bias to higher SMBs than MAR (Fettweis et al., 2020). Notwithstanding this small disagreement, the primary contribution of our study is not the comparison with more complex models, but the fact that the high numerical efficiency of BESSI enables a more comprehensive analysis of model uncertainty, for example by extending the climate model pool to 26. The difference between the highest and lowest SMB in the last simulated

years in our ensemble is more than three times as large as in Hanna et al. (2020) (Fig. 9a).

As in our high emission scenario simulations with BESSI, Fettweis et al. (2013) also find a non-linear SMB decrease in simulations with MAR for the high-end scenario of CMIP5 (RCP8.5) (Fig. 9b). Likewise, the roughly linear trend in the MAR simulations forced by the moderate scenario RCP4.5 is qualitatively analogous to scenario SSP245. The differences between

climate models in the SMB simulations of Fettweis et al. (2013) are comparable to the interquantile range of our study. Another moderate scenario simulation with MAR was performed by Fettweis et al. (2008) for the CMIP3 A1B scenario (Fig. 9b), which





is an intermediate scenario with greenhouse gas emissions between those in SSP245 and SSP370 (Fettweis et al., 2008; O'Neill et al., 2016). It also shows an approximately linear decrease in SMB, however with a smaller uncertainty range than in our moderate SSP245 scenario simulation with BESSI. The multiple regression performed in Fettweis et al. (2008) can reduce the

uncertainty, as non-linear effects are not included there. Additionally, the smaller variations between climate models in CMIP3 compared to CMIP6 can have an effect on the uncertainty of the snow model simulations because of the smaller variability in sensitivity to the carbon dioxide forcing (ECS) (Meehl et al., 2020).

The uncertainty in snow model parameter is negligibly small compared to the other uncertainty components, so that our

results hardly depend on the specific set of parameters in BESSI. However, this does not represent the total uncertainty of SMB modelling, as analysed in Fettweis et al. (2020). To address this question fully, our simulations would have to be repeated with every SMB model of that earlier study. This is not practicable because for some of the SMB models the computational requirements are too high to conduct several hundred simulations. In addition, the total variance of our ensemble is a conservative approximation because our bias correction reduces the variations between the historical simulations of different climate models

and thus also the variability of the climate projections. Furthermore, our assumption of constant topography leads to a bias in SMB projections in 2100 of approximately 10% (Vizcaino, 2014). Finally, climate models do not seem to simulate Greenland blocking correctly (Davini and D'Andrea, 2020). If the frequency of Greenland blocking in summer increases further in a warming climate, our results would be a conservative approximation of the SMB. In spite of these caveats, the substantial difference between the climate model and snow model parameter uncertainties suggests that the climate model uncertainty is

the largest source of error in the future projections of the GrIS SMB.

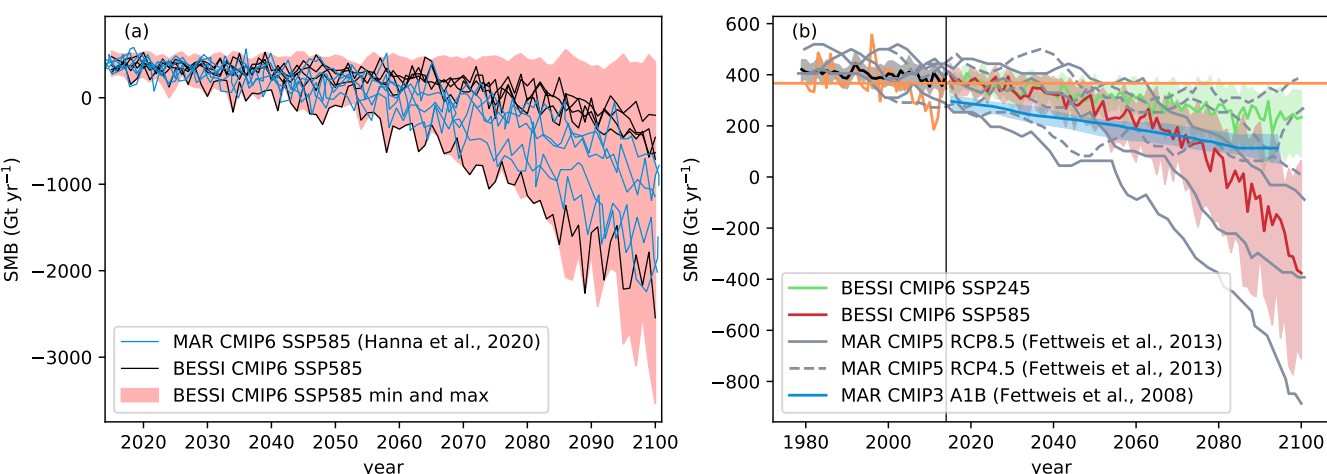

**Figure 9.** (a) SMB simulated by the regional climate model MAR (blue; Hanna et al. (2020), Fig. 11) and the mean of our simulations (black), forced by the same CMIP6 models, scenario SSP585. The red shading illustrates the minimum and the maximum of SMB for our entire ensemble for this scenario. (b) Comparison of our simulations with BESSI, Fettweis et al. (2013), Fig. 4a, and Fettweis et al. (2008), Fig. 7a. Fettweis et al. (2013) uses three different climate models as input, so that there are three grey lines for every scenario. The shading are 25 % and 75 % percentiles.





## Appendix A: Treatment of melted ice in the snow model results

The snow model calculates the SMB for every grid cell on the land surface of Greenland. In the results, only grid cells should be considered which belong to the Greenland ice sheet. The snow model was tuned with the comprehensive RCM RACMO2.3, therefore the RACMO-ice mask (Noël et al., 2016) is used to identify the grid cells with ice. In addition, we restrict the analysis

to grid cells that have an ice thickness of at least 50 m according to ETOPO.

Because we do not simulate ice dynamics, the ice thickness stays constant throughout the simulations with the snow model. For each timestep, BESSI calculates the ice that potentially melts at each grid box, regardless of whether ice is actually present or not. The combination of melt of ice, melt of snow, refreezing, snow, rain and runoff is the mass balance. Therefore, grid

cells with thin ice cover can distort the mass balance, when melt of ice which has already melted is added to the mass balance. This needs to be corrected.

To determine in which grid cells the ice has melted entirely, we subtract the melted ice from the initial ice topography and also consider the inflow by convergence of the lateral steady state flux. If the result is negative, which means that more ice has

melted than would be possible, the grid cell is not considered in the calculation of the mass balance. The ice thickness $\mathrm{d}h$ that is added to each grid cell by ice flux is calculated by the advection equation:

$$\mathrm{d}h = -\operatorname{div}(\boldsymbol{v} \cdot d)\,\mathrm{d}t \tag{A1}$$

where $d$ is the thickness of the ice in the initial topography and $\mathrm{d}t$ is the time step. We use the mean ice velocity $\boldsymbol{v}$ from Nagler et al. (2015) and assume that it is constant. Negative values of $\mathrm{d}h$ are treated as zero for this correction. In grid cells with

thinner ice than a certain threshold, here 50 m, we cannot assume that the ice velocity is constant and therefore we do not take them into account in the SMB calculation.

## Appendix B: Climate models from CMIP6

Several climate models show strong oversaturation in areas with very low temperature, but only small oversaturations can occur in nature due to a lack of freezing nuclei. In climate models, large oversaturations can be caused by e.g. interpolation

from the climate model levels to near-surface output. Some climate modelling groups truncate the relative humidity to 100 % before they make the data available (Ruosteenoja et al., 2017). To obtain physically realistic values, we truncated the relative humidity to 100 % in all climate models used in this study. The climate models HadGEM3-GC31-LL, HadGEM3-GC31-MM and UKESM1-0-LL have a 360 day calendar, thus five days (spread evenly over the year) are taken twice. We used only one ensemble member of each climate model.





**Table B1.** CMIP6-Models (Eyring et al., 2016) used in this project. Data downloaded from https://esgf-node.llnl.gov/search/cmip6/.

| Model | Institution | Grid | DOI |
|---|---|---|---|
| ACCESS-CM2 | Collaboration for Australian Weather and Climate Research | 144x192 | 10.22033/ESGF/CMIP6.4271 |
| | | | 10.22033/ESGF/CMIP6.2285 |
| ACCESS-ESM1-5 | Collaboration for Australian Weather and Climate Research | 145x192 | 10.22033/ESGF/CMIP6.4272 |
| | | | 10.22033/ESGF/CMIP6.2291 |
| BCC-CSM2-MR | Beijing Climate Center | 160x320 | 10.22033/ESGF/CMIP6.2948 |
| | | | 10.22033/ESGF/CMIP6.1732 |
| CanESM5 | Canadian Centre for Climate Modelling and Analysis | 64x128 | 10.22033/ESGF/CMIP6.3610 |
| | | | 10.22033/ESGF/CMIP6.1317 |
| CESM2 | National Center for Atmospheric Research | 192x288 | 10.22033/ESGF/CMIP6.7627 |
| | | | 10.22033/ESGF/CMIP6.2201 |
| CESM2-WACCM | National Center for Atmospheric Research | 192x288 | 10.22033/ESGF/CMIP6.10071 |
| | | | 10.22033/ESGF/CMIP6.10026 |
| CMCC-CM2-SR5 | Euro-Mediterranean Centre on Climate Change | 192x288 | 10.22033/ESGF/CMIP6.3825 |
| | | | 10.22033/ESGF/CMIP6.1365 |
| CNRM-CM6-1 | Centre National de Recherches Météorologiques | 128x256 | 10.22033/ESGF/CMIP6.4066 |
| | | | 10.22033/ESGF/CMIP6.1384 |
| CNRM-ESM2-1 | Centre National de Recherches Météorologiques | 128x256 | 10.22033/ESGF/CMIP6.4068 |
| | | | 10.22033/ESGF/CMIP6.1395 |
| EC-Earth3 | EC-Earth consortium | 256x512 | 10.22033/ESGF/CMIP6.4700 |
| | | | 10.22033/ESGF/CMIP6.251 |
| EC-Earth3-Veg | EC-Earth consortium | 256x512 | 10.22033/ESGF/CMIP6.4706 |
| | | | 10.22033/ESGF/CMIP6.727 |
| FGOALS-g3 | State Key Laboratory of Numerical Modeling for Atmospheric Sciences and Geophysical Fluid Dynamics, Institute of Atmospheric Physics | 80x180 | 10.22033/ESGF/CMIP6.3356 |
| | | | 10.22033/ESGF/CMIP6.2056 |
| GFDL-CM4 | Geophysical Fluid Dynamics Laboratory | 180x288 | 10.22033/ESGF/CMIP6.8594 |
| | | | 10.22033/ESGF/CMIP6.9242 |
| GFDL-ESM4 | Geophysical Fluid Dynamics Laboratory | 180x288 | 10.22033/ESGF/CMIP6.8597 |
| | | | 10.22033/ESGF/CMIP6.1414 |
| HadGEM3-GC31-LL | Hadley Centre for Climate Prediction and Research | 144x192 | 10.22033/ESGF/CMIP6.6109 |
| | | | 10.22033/ESGF/CMIP6.10845 |
| HadGEM3-GC31-MM | Hadley Centre for Climate Prediction and Research | 324x432 | 10.22033/ESGF/CMIP6.6112 |
| | | | 10.22033/ESGF/CMIP6.10846 |
| IPSL-CM6A-LR | Institut Pierre Simon Laplace | 143x144 | 10.22033/ESGF/CMIP6.5195 |
| | | | 10.22033/ESGF/CMIP6.1532 |





| Model | Institution | Grid | DOI |
|---|---|---|---|
| MIROC6 | University of Tokyo, Japan Agency for Marine-Earth Science and Technology | 128x256 | 10.22033/ESGF/CMIP6.5603 |
| | | | 10.22033/ESGF/CMIP6.898 |
| MIROC-ES2L | University of Tokyo, Japan Agency for Marine-Earth Science and Technology | 64x128 | 10.22033/ESGF/CMIP6.5602 |
| | | | 10.22033/ESGF/CMIP6.936 |
| MPI-ESM1-2-LR | Max Planck Institute for Meteorology | 96x192 | 10.22033/ESGF/CMIP6.6595 |
| | | | 10.22033/ESGF/CMIP6.793 |
| MPI-ESM1-2-HR | Max Planck Institute for Meteorology | 192x384 | 10.22033/ESGF/CMIP6.6594 |
| | | | 10.22033/ESGF/CMIP6.2450 |
| MRI-ESM2-0 | Meteorological Research Institute, Japan Meteorological Agency | 160x320 | 10.22033/ESGF/CMIP6.6842 |
| | | | 10.22033/ESGF/CMIP6.638 |
| NESM3 | Nanjing University of Information Science and Technology | 96x192 | 10.22033/ESGF/CMIP6.8769 |
| | | | 10.22033/ESGF/CMIP6.2027 |
| NorESM2-LM | Norwegian Climate Center | 96x144 | 10.22033/ESGF/CMIP6.8036 |
| | | | 10.22033/ESGF/CMIP6.604 |
| NorESM2-MM | Norwegian Climate Center | 192x288 | 10.22033/ESGF/CMIP6.8040 |
| | | | 10.22033/ESGF/CMIP6.608 |
| UKESM1-0-LL | UK Met Office, NERC research centres | 144x192 | 10.22033/ESGF/CMIP6.6113 |
| | | | 10.22033/ESGF/CMIP6.1567 |

## Appendix C:  Uncertainty estimation

To separate the different sources of uncertainty in our projections, we employ the approach by Hawkins and Sutton (2009).
Between the different climate models $M$, scenarios $S$ and perturbed snow model parameters in BESSI $B$, this analysis covers
1952 simulations. Assuming that the running average decadal mean of the simulated SMB $X_{B,M,S,t}$ can be expressed as the
result of these uncertainty contributors and time, as indicated by the subscripts, the snow model output can be divided into a
smooth fit with a forth degree polynomial $P_{B,M,S,t}$ and a deviation $\varepsilon_{B,M,S,t}$ from that fit:

$$X_{B,M,S,t} = P_{B,M,S,t} + \varepsilon_{B,M,S,t}. \tag{C1}$$

We analyse the running average decadal means to facilitate the polynomial fit. The polynomial $P$ can be further divided into a
constant reference SMB $i_M$ that only depends on the climate model, and a deviation $x_{B,M,S,t}$:

$$X_{B,M,S,t} = x_{B,M,S,t} + i_M + \varepsilon_{B,M,S,t}. \tag{C2}$$

We perform the analysis with $x_{B,M,S,t}$ so that we do not have to account for the constant climate model offset. The reference
SMB $i_M$ is the mean of the annual mean values from the time period 1979-2014, averaged over all BESSI configurations.
The spread of the fit matches the spread of the SMB, and the deviations from the fit are only large for few simulations at the





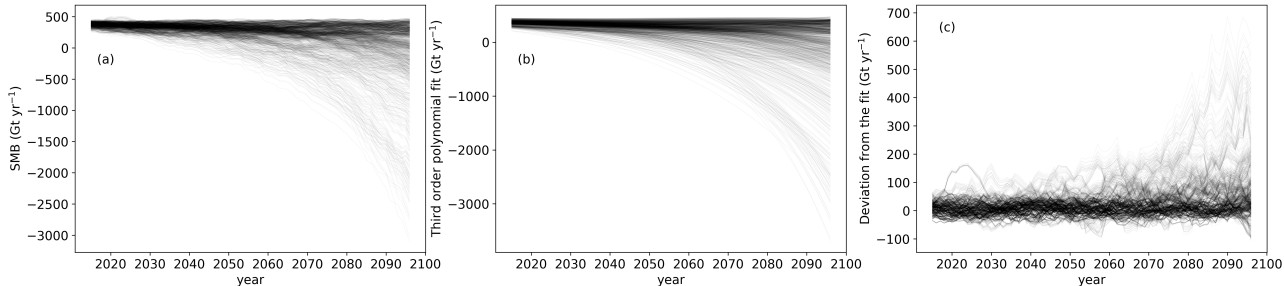

**Figure C1.** (a) Decadal running means of SMB for every parameter-scenario-climate model-combination. (b) Forth degree polynomial fits of the curves in (a). (c) Deviations of the curves in (a) from the fit in (b).

end of the simulated period (Fig. C1). We give more weight to the climate models that perform well in the historical period compared to ERAinterim which we use as a reference. For the calculation of the weights, the average over the SMB of all different parameter combinations for the same climate model is determined first. The absolute deviation of the climate model simulation from ERAinterim is the difference of the mean SMB over the historical period for all parameter combinations:

$\overline{SMB}_{M,79-14} - \overline{SMB}_{E,79-14}$. Additionally, the performance of the climate models is also measured by taking the difference in SMB change over the time period between the climate model and ERAinterim. For every climate model, the total deviation $d_M$ is obtained through the Euclidian distance of the absolute deviation and the deviation of the change:

$$d_M = \sqrt{(\overline{SMB}_{M,79-14} - \overline{SMB}_{E,79-14})^2 + ((\overline{SMB}_{M,04-14} - \overline{SMB}_{E,04-14}) - (\overline{SMB}_{M,79-89} - \overline{SMB}_{E,79-89}))^2}. \quad \text{(C3)}$$

M stands for climate model, E for ERAinterim, and the numbers for the years. The weights are obtained from the deviation

like this:

$$w_M = \frac{1}{d_M} \quad \text{(C4)}$$

The weights are normalised through dividing by their sum, and the normalised weights are denoted $W_M$. The variance of the SMB can be split into components according to the law of total variance. There are 6 possibilities how the split is performed exactly:

$\text{Var}(x) = \text{E}_{S,B}[\text{Var}_M(x|S,B)] + \text{E}_S[\text{Var}_B(\text{E}_M[x|S,B]|S)] + \text{Var}_S(\text{E}_{B,M}[x|S]) \quad \text{(C5)}$

$\text{Var}(x) = \text{E}_{S,B}[\text{Var}_M(x|S,B)] + \text{E}_B[\text{Var}_S(\text{E}_M[x|S,B]|B)] + \text{Var}_B(\text{E}_{S,M}[x|B]) \quad \text{(C6)}$

$\text{Var}(x) = \text{E}_{S,M}[\text{Var}_B(x|S,M)] + \text{E}_S[\text{Var}_M(\text{E}_B[x|S,M]|S)] + \text{Var}_S(\text{E}_{M,B}[x|S]) \quad \text{(C7)}$

$\text{Var}(x) = \text{E}_{S,M}[\text{Var}_B(x|S,M)] + \text{E}_M[\text{Var}_S(\text{E}_B[x|S,M]|M)] + \text{Var}_M(\text{E}_{S,B}[x|M]) \quad \text{(C8)}$

$\text{Var}(x) = \text{E}_{M,B}[\text{Var}_S(x|M,B)] + \text{E}_M[\text{Var}_B(\text{E}_S[x|M,B]|M)] + \text{Var}_M(\text{E}_{S,B}[x|M]) \quad \text{(C9)}$

$\text{Var}(x) = \text{E}_{M,B}[\text{Var}_S(x|M,B)] + \text{E}_B[\text{Var}_M(\text{E}_S[x|M,B]|B)] + \text{Var}_B(\text{E}_{S,M}[x|B]) \quad \text{(C10)}$

The possibilities C8, C9 and C10 are discarded because expectation values of variances between scenarios are calculated. However, we assume that there should be differences between the scenarios because of their different extents of external forc-


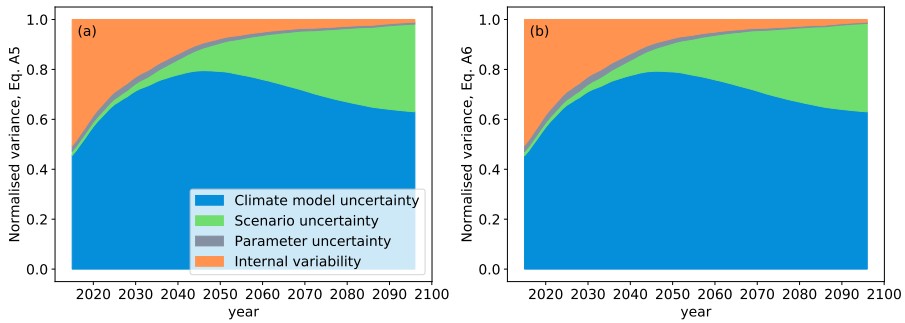

**Figure C2.** Variance components, normalised with the total variance of the fit. (a) Calculated with Eq. C5. (b) Calculated with Eq. C6.

ing. We base our analysis on C7, but the results of C5 and C6 do not deviate much (Fig. 5 and Fig. C2).

The internal variability $V(t)$ is the variance of the residues of the polynomial fit. It is considered time-dependent because the spread between the different simulations in Fig. C1c changes in time. Therefore, it is calculated for every point in time $t$ over the 20 years around $t$ $(t \pm 10a)$ and over all scenarios and BESSI parameters. The weighted mean of this variance over all climate models yields the internal variability:

$$V(t) = \sum_M W_M \text{Var}_{B,S,t \pm 10a}(\varepsilon_{B,M,S,t \pm 10a}). \tag{C11}$$

The sum of the internal variability and the other uncertainty components (Eq. C7) that are considered as the climate model uncertainty $M(t)$, scenario uncertainty $S(t)$ and the snow model parameter uncertainty $B(t)$ is the total variance of the SMB $T(t)$:

$$T(t) = V(t) + M(t) + S(t) + B(t) \tag{C12}$$

For the climate model uncertainty, the weighted variance $\text{Var}_M^w$ of the climate models over the mean parameter configuration
is averaged over the scenarios:

$$M(t) = \text{E}_{S,B}[\text{Var}_M^w(x|S,B)] = \frac{1}{N_s} \sum_s \text{Var}_M^w \left( \frac{1}{N_B} \sum_B x_{B,M,S,t} \right). \tag{C13}$$

For the scenario uncertainty, the variance of the weighted multimodel mean of the mean parameter configuration is taken:

$$S(t) = \text{E}_B[\text{Var}_S(\text{E}_M^w[x|S,B]|B)] = \text{Var}_S \left( \sum_M W_M \left( \frac{1}{N_B} \sum_B x_{B,M,S,t} \right) \right). \tag{C14}$$

The BESSI uncertainty is the mean uncertainty of all parameters:

$$B(t) = \text{Var}_B(\text{E}_{S,M}^w[x|B]) = \frac{1}{N_S} \sum_S \sum_M W_M \text{Var}_B(x_{B,M,S,t}). \tag{C15}$$





*Author contributions.* KMH prepared the model input, conducted the experiments, analysed the results and wrote the main part of the manuscript. TZ prepared the model experiments, applied the statistical methods, reviewed the analysis and revised the manuscript. AB conceived the study, experimental design and analysis, and contributed to the writing of the manuscript.

*Competing interests.* The authors declare that they have no conflict of interest.

*Acknowledgements.* All authors acknowledge support by the Trond Mohn Foundation. KMH received financial support through an Erasmus+ traineeship.



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
