# Peer review of "Sources of Uncertainty in Greenland Surface Mass Balance in the 21st century."

_The Cryosphere, 2021_

## Referee Comment (RC1)

**Review of Holube, Zolles, Born: "Sources of Uncertainty in Greenland Surface Mass Balance in the 21st century." (The Cryosphere Discussion, Paper: tc-2021-128)**

The manuscript of Holube and others simulate with the BErgen Snow SImulator" (BESSI) the response of the Greenland Ice Sheet (GrIS) and its Surface Mass Balance (SMB) under various warming climates scenarios. The scenarios origin from 26 climate models of the CMIP6 archive and represent four different climate scenarios. The required forcing fields of the numerous climate projection and model combinations drive BESSI in the so-called offline modus. Since the SMB of Greenland is the principal driver of the Greenland Ice Sheet's sea-level contribution, it is paramount to probe possible uncertainties of the SMB projections. The authors group the uncertainty estimates into four components:

1. the selected climate scenario that ranges from strong mitigation to business-as-usual,

2. the spread among climate model inherits from its internals that leads to diverting climatic projections under the same climate scenario (item 1),

- parameter uncertainties of BESSI,

- and the residual named internal variability

Since the sea-level contributions of the global ice sheets, such as Greenland, are still highly uncertain, this manuscript addresses an important issue. It has been a pleasure to read the well-structured manuscript, but I also confident that improvement of the language could help many readers to enjoy the work besides essential corrections.

**I recommend the publication of the manuscript after a minor revision.**

In the following, I will first discuss general concerns before I raise more specific and technical issues.

**General comments**

As part of your uncertainty estimates, you are using those combinations of parameters that reproduce the SMB of the reference models RACMO. What would be needed to restrict the valid parameter space? Since you also compute the exchange of fluxes between the surface and the atmosphere within BESSI, would comparing the internally computed fluxes with references lead to a restriction of the uncertainty? Although the flux estimates coming from different regional climate models, such as RACMO, MAR, and HIRHAM, would probably differ, would you please explain a bit what could be done to reduce the uncertainties if appropriate observations could be made available? To

In some figures, the unit of matter fluxes (precipitation) or radiation fluxes (shortwave and longwave) are usually per time unit; otherwise, the numbers might be wrong. Please check, for example, Figures 1, 3, 6, 7, while the mass flux of Figures 2, 9 seem to be correct. Please also check the related text.

Since my mother tongue is not English, I know it can be hard to write English, which is nice to read and understandable. Nevertheless, I've gotten the impression that the manuscript would gain vigor with an improved language. In the more specific issue section, I provide examples for some cases. However, the list is not exhaustive.

**Specific issues**

In the following, the abbreviations "L" or "l" stands for the line number of the manuscript, and "p" and "P" for page.

L1 (abstract): Please replace "that" with "which" because it is not a defining clause; see https://www.grammarly.com/blog/which-vs-that/, for instance

L61: Since much effort is put into preparing BEDMAP, for instance, how significant are the differences between ETOPO and BedMachine v3 (Morlighem et al., 2017), for example, and how sensitive are your results considering the applied height corrections?

L62: Can you please be so kind and indicate how the results would change if you use instead of daily, sub-daily forcing reproducing the daily cycle.

L67: Could you please elaborate on how the simplified horizontal mass flux impacts the results compared to entirely ignoring the flux versus a fully dynamical ice sheet model?

You might add a sentence like: "The uncertainty related to the simplified representation of the ice flow is not addressed further."

L74-75: BESSI is compared to RACMO. What went into the comparison? The integrated SMB, spatial SMB field, or even the different fields contributing to the SMB, such as precipitation, melting, refreezing, sublimation.

L81: I heard about the Pareto optimality in the relationship of economic science. Since this might not be commonly known, would you please be so kind as to add one sentence summarizing its basic concept?

P3, L71-84: I guess you only vary the following parameters: fresh snow albedo, firn

albedo, and turbulent heat exchange coefficient. Please state clearly, by adding a sentence like: "The combination of these parameters contribute to the parameter uncertainty discusses below."

P3, L71-84: You may add: "The albedo of bare ice is fixed with a value of 0.35."

P5, L91-95: The property difference between the historical and future period is added to the reference climate property from ERAinterim. Do you use the entire historical period (1850-2014)? Would you please clarify the text?

P5, L94-95: I guess you perform the computation for each calendar day independently? How do HOW DO YOU FILL THE MISSING DAYS IF THE MODEL HAS ONLY 360 DAYS?

L104-105: Not sure, but have you swap shortwave and longwave in the text or Figure 1d+1e?

L106-107: I do not fully understand the second half of the sentence. In particular, what is meant by "model differences overlap most for different scenarios." Please clarify.

L109-110: I understand these two sentences, but initially, I have not. Would it be possible to improve them?

L 112: You mention 96 combinations, which is not the multiple of 26 climate models. I guess that some combinations of models and scenarios are not available. I suggest adding to the caption of Table B1: "For each of the listed models, we use the scenarios SSP126, SSP245, SSP370, SPP585 to drive BESSI; except for some missing model scenario combinations. XXX misses SSPxxx, YYY: SSPyyy, ZZZ: SSPzzz1 and SSPzzz2, … ." and expend the sentence of line 112: ".... 96 selected climate model-scenario combinations (Table B1)."

L120: How fast does the bias in the identical starting conditions, which may not be consistent with all parameter combinations, disappear?

L126: You state that a small amount of daily precipitation is unrealistic in the North. Could you add a reference to confirm this statement?

L130: You may replace "..., but the monthly averages are the same" by "..., but the monthly averages are identical" or "..., but the monthly averages are similar."

L131: I guess "Therefore" is not appropriate than "thus."

L133: You may help the reader to list the transient variables shortly by writing, for example:"... with different transit variables (air and dew point temperature, precipitation, short and longwave radiation)."

L141-143: You compare the "range in simulated SMB" with the "range in input variable" in terms of "magnitude." Since these do not necessarily have the same unit, improve the ambiguous wording to avoid comparing apples and oranges.

L149-150: Here, you describe the models' inability to reproduce atmospheric blocking. Since the horizontal resolution of most models (Table B1) prevents a representation, you may modify the sentence:"... (Fig.2, black) because the coarse horizontal resolution

hampers the representation of the observed blocking and its increased activity (Davini and D'Andrea, 2020)."

L155-156: I suggest: "There, heavier precipitation occurs under a warmer climate."

L156: Please drop the awkward wording "greatly" or replace it.

L157-159: Please avoid greater, and the appearance of the information in the parentheses is confusing; you may turn it into an actual sentence. I suggest:" These SMB changes are much more pronounced in the high-end scenario SSP585 because of the enhanced/amplified/more robust change in the input variables."

L159: In the sentences above, you describe simulation results and now compare them with observational estimates. Please make this distinction clear, for example,": Currently observed SMB changes are dominated by amplified melting … ." Also, check the manuscript if you use termination "melting" versus "melt."

L161: You have found a 6°C warming across Greenland in SPP585. Just for curiosity, how strong is the warming in the altitude of the ablation zones, such as the altitude up to 1500 m?

L162-164: I find this sentence not clear enough, and, I guess, you talk about the actual equilibrium line that spreads across a larger area (Fig. 4) for more substantial warming. Therefore, please sharpen the sentence. I suggest: "Since ice sheet margin experiences the highest melting rates, its relative standard deviation reaches highest values near/along the actual(?) equilibrium line. Therefore, the choice of the climate model is decisive for the SMB in this region."

L164: Here, you say, "the equilibrium line varies substantially more." Since the title of your manuscript refers to uncertainty, you may rephrase "the equilibrium line position is subject to substantial uncertain."

L166: Please clarify if you mean: "... the differences between climate models driven by the same scenario increases with stronger greenhouse gas forcing (Fig. 2)."

P9, L168-175: This paragraph should be sharpened.

L169-170: There, you make the assumption that you can split the uncertainty into four components. I guess you further assume that these components are independent. Therefore I suggest rephrasing: "We assume that the total uncertainty of the simulations could be split/separated into four independent components: climate model, climate scenario to drive a chosen climate model, BESSI parameter uncertainties (albedo of fresh snow and firn, turbulent heat exchange coefficient), and internal variability."

L170: Please add a sentence stating what internal variability is or prove an example.

L172: Do you mean:" ... the decadal running mean of the SMB at each grid point ..."?

L173: Replace "are" buy "is."

L177: You do not really talk about visibility. I suggest:"... the different uncertainty components can be clearly identified when normalised with the sum... ."

L190: Do you mean here, "This pronounced uncertainty is larger than the differences"?

L190: Unclear, what do you mean by "the difference of the model consensus between the scenarios"? Please clarify.

L191-192: The sum of all components is 99% and not 100%. Please correct.

L195: I'm unsure if "the SMB changes most" is appropriate. Would you mind checking and rephrasing if applicable?

L208: Please rephrase:"...variables on the SMB across the entire GrIS and three regions previously used by Zolles and Born (2019)(Fig. 7).

L219-222: You may rephrase and split the sentence. I suggest:"... simulation (Fig. 7) driven by the SSP585 scenario. It highlights non-linearities that amplify the SMI reduction. For example, air temperature and precipitation often covary so that the increased precipitation compensates the increased melt only partly. If heavier precipitation delivers more rain, the energy required to refreeze the additional rain in the snowpack increases its heat."

L222-224: Here, you create the impression that the combination of a transient air temperature and longwave radiation explains the non-linearities. I believe you, but you have performed only simulations where one variable is transient and not two, or? If this is the case, please formulate it more carefully. You may add at the beginning of the sentence: We conclude that when the air temperature and longwave ….".

L223: Please be very clear about the distinction between observation deduced from measurements and simulated results. Hence, replace "is observed" with "is detected."

L225: Please clarify if you mean: "on the vertical temperature gradient in the snow."

L226-228: Do you mean: "Since the sublimation is driven by the saturation pressure difference between lower atmosphere and surface, sublimation increased for a higher dew point temperature while it is reduced for a warmer surface."

L226-228: Unclear sentence. Please improve.

L243-244: You may shorten: "As a consequence, the scenario uncertainty is reduced (Fig. 8b)."

L274: Since you talk about the unknown future, I would like to suggest a slight change:" temperature changes are probably exacerbated."

L301-302: A sentence is missing stating that increased blocking leads to amplified mass loss. Would you please add an appropriate citation?

L303: I suggest rephrasing: "Therefore, our future SMB projections are conservative because the climate models do not fully represent the expected increase of the Greenland block in a warming climate."

**Appendix**

L327: I suggest rephrasing: "show strong oversaturation of humidity in areas with very low temperatures while only small oversaturation occurs in nature... ."

L342: Please add "t":" of these uncertainties contributors and time $t$ as indicated … ."

**Figures**

The figures are in general of high quality and well prepare. However, some technical issues remain, which are discussed below.

Some figures having a time axis do not show the evolution until 2100 (Figure 5, 8). Is this an artifact? If so, please correct it.

Figure 1: Please state the meaning of the vertical line in all subplots. Would it be possible to indicate the used EraInterim data period in the plots? Are these line plots the actual medians, or are these the climate model anomalies relative to their related historical or pre-industrial climate states? If so, please clarify. In the caption, you may state that the radiative fluxes are "at the surface." Since you use for the first time the unit "kg m$^{-2}$" for precipitation, you may add the following note: "Please note the precipitation unit, 1 kg m$^{-2}$ equals 1 mm(WE); WE=Water equivalent."

Figure 3: The labels indicating the subplot (a-d) are below the black ground. Please repair.

Would you please move the column labels above the top figure row? The subplot in the lowest row shows only black and grey patterns and does not resemble the values of the related colorbar. Please fix. In the figure caption, I consider the last sentence using the wording "meaningful" as jargon. Please improve.

Figure 5: Great figure, but would it be more appropriate to represent the square root of the variance/standard deviation in subplot (a)? In this case, the values could be directly compared to recent mass balance estimates and sea-level potential. If you would like to keep your variance y-axis, you may add a second axis with the square root values.

Figure 6: The subplot labels are cover by the black subfigure's background. Please fix. As mentioned for Figure 5, would it be possible to present the square root of the variance too? Why do you not use for the right column the last complete decadal period 2091-2100?

Figure 7: In the subplot (e), only the region "East" is shown. Please repair.

**Table**

Table 1: Please separate the units corrected by replacing "kgm$^{-2}$" with "kg m$^{-2}$", for instance.

**Bibliography**

Morlighem, M., Williams, C. N., Rignot, E., An, L., Arndt, J. E., Bamber, J. L., Catania, G., Chauché, N., Dowdeswell, J. A., Dorschel, B., Fenty, I., Hogan, K., Howat, I., Hubbard, A., Jakobsson, M., Jordan, T. M., Kjeldsen, K. K., Millan, R., Mayer, L., Mouginot, J., Noël, B. P. Y., O'Cofaigh, C., Palmer, S., Rysgaard, S., Seroussi, H., Siegert, M. J., Slabon, P., Straneo, F., van den Broeke, M. R., Weinrebe, W., Wood, M. and Zinglersen, K. B.: BedMachine v3: Complete Bed Topography and Ocean Bathymetry Mapping of Greenland From Multibeam Echo Sounding Combined With Mass Conservation, Geophys. Res. Lett., 44(21), 11,051-11,061, doi:10.1002/2017GL074954, 2017.

---

## Referee Comment (RC2)

**Sources of uncertainty in Greenland surface mass balance in the 21st century**

K. M. Holube, T. Zolles, and A. Born

**Summary**

The authors utilize an offline energy and mass balance model forced with CMIP6 global climate model simulations to estimate uncertainty in projected Greenland ice sheet surface mass balance for the 21st century. The authors assess the impact of discrepancies between climate models, differences between future scenarios, and internal SMB model uncertainty in the projections. They find that the largest uncertainty results from differences between climate models, followed by scenario uncertainty, followed by snow model parameter uncertainty.

**General Comments**

The manuscript is well written and well laid-out. I find the authors' approach to be overall logical. I believe the manuscript is well suited for the cryosphere and should be accepted. However, I have some general and specific points that I think the authors should address before the manuscript can be published:

(1) I think the authors should provide some additional detail about the parameterizations used in BESSI and whether the parameter changes used in the BESSI sensitivity studies really representative of the uncertainty in modeling the Greenland snow and ice surface. I would imagine that comparing multiple SMB models across simulations would add to uncertainty in this component. There are processes (e.g. the evolution of bare ice albedo) that are still not well understood or included in models, and which could add uncertainty in future projections. The authors should discuss these potential caveats in further detail especially with regard to their conclusion that the uncertainty associated with the snow model parameters is small.

(2) The authors utilize a single ensemble member from each GCM to evaluate the inter-model uncertainty. The inter-model uncertainty is therefore influenced to some extent by the internal variability of each model. The question this raises is whether the inter-model variability is larger than the internal variability of any one model. The authors' assumption seems to be that the spread of the single ensemble members from each model is indicative of the uncertainty caused by the inter-model uncertainty. It is not clear whether this is the case. For a more definitive result, it would be useful to perform some additional experiments: (1) Forcing BESSI with the ensemble mean from each simulation rather than single ensemble members, and (2) Forcing BESSI with multiple ensemble members from a single model to compare the inter-model vs. single-model spread. It would be useful to have at least one of these additional simulations if the authors think this makes sense.

(3) The authors note that they linearly interpolate GCM input variables onto the 10 km BESSI grid.  This seems somewhat problematic, especially in coastal areas where there is a high degree of spatial variability.  Because temperature, for example, is dependent on elevation, downscaling methods are often employed to take this into account (e.g. Noël et al...., Fischer et al., 2014).  The simple linear interpolation seems likely to lead to biases in the SMB forcing fields.  However, as the authors are looking at differences on a broad scale, it might be less important.  The authors should discuss the impact that this might have on the results, and if possible test the impact of a different more sophisticated downscaling technique to evaluate the impact on the results.

(4) The authors frequently refer to General Circulation Model/Global Climate Model/ Earth System Models simulations as "climate models".  However, Regional Climate Models are also "climate models", and the use of the term "climate models" is sometimes confusing.  I suggest replacing the term "climate models" with "GCMs" to make clear that these are global simulations.

**Specific Comments**

1. **Line 17:** Add "currently" after "(GrIS)" for clarity.
2. **Lines 27-35:** Here, following the first sentence, the authors introduce what is done in this study, but then go back to describing previous evaluations in the next paragraph.  BESSI is also mentioned in this paragraph, but then introduced later, in the last paragraph of the introduction.  I suggest moving this material in this paragraph and combining it with the last paragraph of the introduction, and making clear the different sources of uncertainty that are being addressed.
3. **Line 37:** Define the term PDD here.
4. **Lines 41-42:** Here the authors should make it clear that RCM simulations are used to dynamically downscale GCM simulations, which often do not have the spatial resolution or detailed physical representation of the ice sheet surface needed to simulate SMB reasonably well.
5. **Line 43:** I suggest changing "evaluating" to "downscaling", after clarifying the reason for using RCMs in the previous sentence.
6. **Lines 44-46:** Here the sentence is a bit unclear.  Suggested correction:  "In Fettweis et al. (2008), which utilized RCM simulations to project SMB forced with a subset of CMIP3 simulations, a multiple regression for the SMB changes as a function of temperature and precipitation is performed to calculate the SMB changes for CMIP3 simulations that were not used to force the RCM."
7. **Lines 52-56:** In addition to adding in material from the second paragraph of the introduction, I would suggest briefly explaining what BESSI is and why it is advantageous in this situation as compared with GCM or RCM simulations.

8. **Lines 61-62:** Please explain how the layers are adjusted. Are the authors referring to the snowfall amount within the grid cell? Is there a maximum thickness of the snow model? If the simulation were run continuously, areas of net positive SMB the amount of snow represented in the model would continuously increase.

9. **Line 62:** Suggest changing to "15 snow or ice layers" for clarity.

10. **Lines 68-69:** What was the result of this comparison?

11. **Line 71**: Can the authors briefly describe how these parameterizations work?

12. **Line 74:** "ERAinterim" should be changed to "ERA-Interim" here and throughout, and a brief description and reference should be added.

13. **Lines 83-84:** Can the authors briefly explain the Bougamont et al. (2005) albedo routine here or at the start of the paragraph? How is the snow vs. ice albedo treated? This is particularly important because the contrast between bare ice and snow albedo plays an important role in the GrIS energy and mass balance (e.g. Ryan et al., 2019).

    Ryan, J. C., Smith, L. C., Van As, D., Cooley, S. W., Cooper, M. G., Pitcher, L. H., & Hubbard, A. (2019). Greenland Ice Sheet surface melt amplified by snowline migration and bare ice exposure. Science Advances, 5(3), eaav3738.

14. **Lines 89-90:** Suggest changing "whereas the dewpoint is calculated…" to "with the exception of the dew point, which is calculated…"

15. **Line 100:** Add "to perform bias correction" after "ratio of the monthly means" for clarity.

16. **Lines 104-105:** From Figure 1 it seems that shortwave radiation increases, while longwave radiation decreases, contrary to what is said here. (Also see note below about the axis label on Fig. 1e.) Could the authors be referring to the SSP126 simulation? If so it might make more sense to discuss all the simulations shown on the figure. Please clarify and/or revise.

17. **Figure 1:** The axis labels on Fig. 1e seem to be incorrect. I would expect the downwelling SW radiation to have a positive value, and to be of the same order of magnitude as downwelling LW radiation. Also change "Temperature in 2 m" to "Temperature at 2 m" and "Dewpoint in 2 m" to "Dewpoint at 2 m" in the caption.

18. **Line 105:** Suggest changing "the larger the increase" to "the larger the change", as there is a larger change in the higher GHG forcing scenarios regardless of the direction of the change.

19. **Lines 105-107:** Rather, it seems that precipitation shows the least overlap between scenarios, as the model spread is small relative to the scenario spread in that case. Please clarify or revise.

20. **Line 128:** What is meant by the "actual" climate simulation? Suggest changing to read "following the temporal distribution of precipitation in the GCM simulation."

21. **Line 133:** Note which variables were changed here.

22. **Lines 145-147:** This is a bit confusing. I suggest revising to note that the effect of the larger change in the input variables is a larger cumulative effect on SMB.
23. **Line 147:** Change "snow model" to "BESSI" for clarity.
24. **Line 158:** Change "increase of" to "change in".
25. **Lines 157-158:** As mentioned above, I'm confused about the SW and LW radiation changes shown on Fig. 1. The axis labels for SW radiation seem to be incorrect. An increase in downward SW radiation would be consistent with the decrease in downward LW radiation that is shown. However, an increase in precipitation might be indicative of increased cloud cover, which would increase downward LW radiation and reduce SW radiation. Please revise the figure and/or the text.
26. **Line 158:** The SW radiation having little effect on SMB seems contrary to recent studies that suggest recent changes in atmospheric circulation lead to increased downwelling SW radiation and increased melt (e.g. Tedesco et al., 2016; Hofer et al., 2017). However, it could be that the combination of factors and feedbacks, which are not included in these idealized experiments, may play an important role in that case. This could be mentioned in section 3.3. Perhaps it should also be clarified here that SW radiation alone does not influence SMB in the idealized experiments performed.
27. **Line 172:** Is this the decadal running mean of SMB for the entire ensemble? Please clarify.
28. **Line 206:** Is this a seasonally varying mean?
29. **Line 241:** Could the authors explain this a bit further? Is this because bare ice is exposed at the surface, and do model assumptions about the snow/ice profile play a role in this feedback? I suppose this effect may fall under the model parameter uncertainty experiments.
30. **Line 289:** Explain the "multiple regression" a bit further.
31. **Figure 9:** The plots here are a bit confusing because the shading on (a) shows the maximum range, while (b) shows the mean and 25th and 75th percentiles. Would it be possible to also shade the 25-75 range a slightly different color and show the mean on (a)? Or to show the maximum range on (b)?
32. **Line 310:** Please define ETOPO and provide a reference.
33. **Line 325:** Can the authors explain the choice of the 50 m threshold? Is it estimated based on previous assessments?

**Technical Corrections**
1. **Line 22:** Change "are met" to "were met".
2. **Line 43:** Suggest changing "leaving their use to" to "leaving their use limited to".
3. **Line 74:** ERAinterim should be changed to ERA-Interim
4. **Line 94:** Change "in the period of 1979-2014" to "over the 1979-2014 period".

5. **Lines 109-111:** Suggest adding "(1)" before "The main ensemble…", and "(2)" before "The 'single forcing'…" for clarity.
6. **Line 161:** Add "there" after "melt increases considerably" for clarity.
7. **Line 170:** Suggest changing to "climate model uncertainty, climate scenario uncertainty, snow model parameter uncertainty, and internal variability."
8. **Line 175:** Change "forth" to "fourth".
9. **Lines 202-203:** Suggest changing to read: "The scenario uncertainty has a similar magnitude as the climate model uncertainty only at the margins of the ice sheet and in the area where the total variance is low."
10. **Line 213:** I believe the correct term is "desublimation" or "deposition".

---

## Author Response (AR1)

**Response to Anonymous Referee #1**

We would like to thank the Anonymous Referee #1 for the detailed comments.

**General comments**

As part of your uncertainty estimates, you are using those combinations of parameters that reproduce the SMB of the reference models RACMO. What would be needed to restrict the valid parameter space? Since you also compute the exchange of fluxes between the surface and the atmosphere within BESSI, would comparing the internally computed fluxes with references lead to a restriction of the uncertainty? Although the flux estimates coming from different regional climate models, such as RACMO, MAR, and HIRHAM, would probably differ, would you please explain a bit what could be done to reduce the uncertainties if appropriate observations could be made available? To

We agree that observational data could improve the parameters of the snow model. However, we would like to keep the discussion of the parameter tuning short because it is subject of the work by Zolles and Born (2021). A detailed discussion of the BESSI parameter uncertainty could distract from the primary goal of this study, to quantify the uncertainty arising from different plausible climate simulations. In line 93-95, we have pointed out that multi-variate optimization always leads to several optimal solutions (Zolles et al. 2019).

In some figures, the unit of matter fluxes (precipitation) or radiation fluxes (shortwave and longwave) are usually per time unit; otherwise, the numbers might be wrong. Please check, for example, Figures 1, 3, 6, 7, while the mass flux of Figures 2, 9 seem to be correct. Please also check the related text.

Thank you for pointing out this mistake. We have corrected the figures by adding the time unit to Fig. 1, 3, 6 and 7. In the process, we realised that the subplots in Fig. 1 were incorrectly labeled, for which we apologise. We have corrected these labels.

Since my mother tongue is not English, I know it can be hard to write English, which is nice to read and understandable. Nevertheless, I've gotten the impression that the manuscript would gain vigor with an improved language. In the more specific issue section, I provide examples for some cases. However, the list is not exhaustive.

**Specific issues**

In the following, the abbreviations "L" or "l" stands for the line number of the manuscript, and "p" and "P" for page.

L1 (abstract): Please replace "that" with "which" because it is not a defining clause; see https://www.grammarly.com/blog/which-vs-that/, for instance

Done

L61: Since much effort is put into preparing BEDMAP, for instance, how significant are the differences between ETOPO and BedMachine v3 (Morlighem et al., 2017), for example, and how sensitive are your results considering the applied height corrections?

Thank you for suggesting to use a more elaborate topography. However, the bed topography is not relevant in our study because we do not run an ice sheet model. Thus, we have compared the topography used in BESSI (ETOPO) with the more recent surface topography reconstruction by Schaffer et al. (2016) in the figure below, using the same ice mask as in the manuscript.

Difference Schaffer et al. - ETOPO

[Figure]

Note the irregular spacing of the colorbars. Temperature differences are calculated from height differences using the moist adiabatic lapse rate (6.5 K/km). Thus, in most grid cells belonging to the ice sheet, the differences between ETOPO and the topography from Schaffer et al. (2016) are negligible. The differences are largest at the margin, where positive as well as negative differences are found, which partially cancel each other out when considering the entire ice sheet. We expect our key results to change less than the figure implies, because they are based on relative differences.

L62: Can you please be so kind and indicate how the results would change if you use instead of daily, sub-daily forcing reproducing the daily cycle.

We agree that neglecting the diurnal cycle may impact processes that can change the SMB. Krebs-Kanzow et al. (2018) found different qualitative changes in the melt rate depending on the length of the melt period with their energy balance model (dEBM). In addition, omitting the diurnal cycle can lead to an underestimation of refreezing (Krebs-Kanzow et al., 2021). However, using a time step of one day is a deliberate choice for BESSI because it greatly improves the computational speed and thereby enables studies such as ours. Since we cannot reliably estimate how the diurnal cycle would affect our results, we have added this caveat to the discussion (l. 351-352).

L67: Could you please elaborate on how the simplified horizontal mass flux impacts the results compared to entirely ignoring the flux versus a fully dynamical ice sheet model?

You might add a sentence like: "The uncertainty related to the simplified representation of the ice flow is not addressed further."

We have added a paragraph on this topic at the end of Appendix A. We believe that for a more realistic calculation, incorporating the melt-elevation feedback would have a greater effect on the SMB than improving the representation of the ice flow, at least for simulations that do not go beyond the end of this century such as ours. The figure below illustrates the temperature change caused by melt of ice in two climate scenarios.

[Figure]

L74-75: BESSI is compared to RACMO. What went into the comparison? The integrated SMB, spatial SMB field, or even the different fields contributing to the SMB, such as precipitation, melting, refreezing, sublimation.

The integrated SMB and the spatial SMB field went into the comparison (l. 87-93). We have clarified that BESSI's performance is compared to the RACMO SMB (l. 84 and 87).

L81: I heard about the Pareto optimality in the relationship of economic science. Since this might not be commonly known, would you please be so kind as to add one sentence summarizing its basic concept?

We have explained our approach in more detail (l. 93-97).

P3, L71-84: I guess you only vary the following parameters: fresh snow albedo, firn albedo, and turbulent heat exchange coefficient. Please state clearly, by adding a sentence like: "The combination of these parameters contribute to the parameter uncertainty discusses below."

When incorporating a comment of Anonymous Referee #2, we have added a paragraph about the parameters used in BESSI (l. 76-82). There we clarify that we vary the albedo and turbulent heat exchange parameters and that these contribute to the parameter uncertainty (l. 76-77).

P3, L71-84: You may add: "The albedo of bare ice is fixed with a value of 0.35."

We have used an ice albedo of 0.4, and added this information in line 99.

P5, L91-95: The property difference between the historical and future period is added to the reference climate property from ERAinterim. Do you use the entire historical period (1850-2014)? Would you please clarify the text?

We have stated that only the time period is used in that both datasets are available (l. 109).

P5, L94-95: I guess you perform the computation for each calendar day independently? How do HOW DO YOU FILL THE MISSING DAYS IF THE MODEL HAS ONLY 360 DAYS?

Appendix B specifies that if the model has only 360 days, five days, spread evenly over the year, are taken twice. For this, we used a nearest neighbor interpolation method between the time axes with different numbers of days.

L104-105: Not sure, but have you swap shortwave and longwave in the text or Figure 1d+1e?

We apologise for a mistake in the labels of Fig. 1, where the precipitation was labeled "dew point",; longwave radiation: "precipitation"; shortwave radiation: "longwave radiation"; and dew point: "shortwave radiation". This has now been corrected.

L106-107: I do not fully understand the second half of the sentence. In particular, what is meant by "model differences overlap most for different scenarios." Please clarify.

We have rephrased the sentence, hinting at the relatively large ranges of precipitation between the climate models compared to the trend over the 21$^{st}$ century (l. 125-127).

L109-110: I understand these two sentences, but initially, I have not. Would it be possible to improve them?

We have rephrased it (l. 129-130).

L 112: You mention 96 combinations, which is not the multiple of 26 climate models. I guess that some combinations of models and scenarios are not available. I suggest adding to the caption of Table B1: "For each of the listed models, we use the scenarios SSP126, SSP245, SSP370, SPP585 to drive BESSI; except for some missing model scenario combinations. XXX misses SSPxxx, YYY: SSPyyy, ZZZ: SSPzzz1 and SSPzzz2, … ." and expend the sentence of line 112: ".... 96 selected climate model-scenario combinations (Table B1)."

Done

L120: How fast does the bias in the identical starting conditions, which may not be consistent with all parameter combinations, disappear?

We conducted the spin up for every parameter combination. The resulting time series of the integrated SMB is shown in the upper panel of the figure below, where the parameter set used in this study is marked red, and the standard deviation of the SMB between the parameter sets is shown in the lower figure.

[Figure]

The temporal mean of the spatially integrated SMB differs by less than 50 Gt/yr between the parameter combination with minimum and maximum SMB. The last 36 years of the spin up are evaluated because there the forcing is ERA-interim reanalysis data from 1979-2014. Fig. 2 of the manuscript shows that changes in SMB of 50 Gt/yr occur in many simulations already a few years after the beginning of the simulation. Furthermore, the SMB differences caused by the parameter combinations affect only few grid cells.

We have added to the manuscript that the SMB bias in the identical starting conditions is generally overcompensated after a few years of climate forcing (l. 141-142).

L126: You state that a small amount of daily precipitation is unrealistic in the North. Could you add a reference to confirm this statement?

We have added a reference to Sodemann et al. (2008), who showed that precipitation in Greenland depends on the NAO.

L130: You may replace "..., but the monthly averages are the same" by "..., but the monthly averages are identical" or "..., but the monthly averages are similar."

Done

L131: I guess "Therefore" is not appropriate than "thus."

Done

L133: You may help the reader to list the transient variables shortly by writing, for example:"... with different transit variables (air and dew point temperature, precipitation, short and longwave radiation)."

Done

L141-143: You compare the "range in simulated SMB" with the "range in input variable" in terms of "magnitude." Since these do not necessarily have the same unit, improve the ambiguous wording to avoid comparing apples and oranges.

We have rephrased the sentence to make it clearer that we compare the ranges of values between climate scenarios (l. 164-166).

L149-150: Here, you describe the models' inability to reproduce atmospheric blocking. Since the horizontal resolution of most models (Table B1) prevents a representation, you may modify the sentence:"... (Fig.2, black) because the coarse horizontal resolution hampers the representation of the observed blocking and its increased activity (Davini and D'Andrea, 2020)."

Done

L155-156: I suggest: "There, heavier precipitation occurs under a warmer climate."

Done

L156: Please drop the awkward wording "greatly" or replace it.

Dropped

L157-159: Please avoid greater, and the appearance of the information in the parentheses is confusing; you may turn it into an actual sentence. I suggest:" These SMB changes are much more pronounced in the high-end scenario SSP585 because of the enhanced/amplified/more robust change in the input variables."

We have adopted the suggested phrasing of the sentence (l. 180-181), and removed the information about shortwave radiation, because we have discussed it in Sect. 3.3 (l. 253-257).

L159: In the sentences above, you describe simulation results and now compare them with observational estimates. Please make this distinction clear, for example,": Currently observed SMB changes are dominated by amplified melting … ." Also, check the manuscript if you use termination "melting" versus "melt."

We have adopted the suggestion (l. 181), and changed "melt" to "melting" in line 249, too.

L161: You have found a 6°C warming across Greenland in SPP585. Just for curiosity, how strong is the warming in the altitude of the ablation zones, such as the altitude up to 1500 m?

At altitudes up to 1500 m, the warming between 2015 and 2100 is about 8 K in SSP585 in the global climate model median.

L162-164: I find this sentence not clear enough, and, I guess, you talk about the actual equilibrium line that spreads across a larger area (Fig. 4) for more substantial warming. Therefore, please sharpen the sentence. I suggest: "Since ice sheet margin experiences the highest melting rates, its relative standard deviation reaches highest values near/along the actual(?) equilibrium line. Therefore, the choice of the climate model is decisive for the SMB in this region."

We thank the Referee for pointing out that this sentence is unclear. We have adopted the suggestion (l. 184-185), but dropped the half sentence about the melting rates because, simply, the relative standard deviation is highest where the SMB is close to zero.

L164: Here, you say, "the equilibrium line varies substantially more." Since the title of your manuscript refers to uncertainty, you may rephrase "the equilibrium line position is subject to substantial uncertainty."

Done

L166: Please clarify if you mean: "... the differences between climate models driven by the same scenario increases with stronger greenhouse gas forcing (Fig. 2)."

Done

P9, L168-175: This paragraph should be sharpened.

L169-170: There, you make the assumption that you can split the uncertainty into four components. I guess you further assume that these components are independent. Therefore I suggest rephrasing: "We assume that the total uncertainty of the simulations could be split/separated into four independent components: climate model, climate scenario to drive a chosen climate model, BESSI parameter uncertainties (albedo of fresh snow and firn, turbulent heat exchange coefficient), and internal variability."

We have rephrased the paragraph to clarify our procedure.

L170: Please add a sentence stating what internal variability is or prove an example.

We have rephrased the paragraph making the assumptions clear and providing an example for internal variability (l. 195).

L172: Do you mean:" ... the decadal running mean of the SMB at each grid point ..."?

We thank the Referee for pointing out that this is unclear. We have clarified that the decadal running mean of the spatially integrated SMB is considered here. The analysis for each grid point is discussed later.

L173: Replace "are" buy "is."

The relative clause does not exist any more after re-formulating the paragraph.

L177: You do not really talk about visibility. I suggest:"... the different uncertainty components can be clearly identified when normalised with the sum... ."

Thank you for pointing out that "visible" is an awkward wording. We have clarified the sentence using a different phrasing (l. 201).

L190: Do you mean here, "This pronounced uncertainty is larger than the differences"?

Yes, we have changed this.

L190: Unclear, what do you mean by "the difference of the model consensus between the scenarios"? Please clarify.

With "model consensus", we meant the median SMB over all GCMs for each climate scenario. We have clarified this (l. 225-226).

L191-192: The sum of all components is 99% and not 100%. Please correct.

Thank you for suggesting to check the sum of the components. However, the sum of shares of GCM uncertainty (62%), climate scenario uncertainty (35%) and the combination of snow model parameter uncertainty and internal variability (3%) is 100%.

L195: I'm unsure if "the SMB changes most" is appropriate. Would you mind checking and rephrasing if applicable?

[Figure]

Absolute SMB difference: Last decade of 21st century-ERAinterim reference period, median over GCMs and snow model parameters.

[Figure]

The figure shows that the largest SMB changes are found close to the margins, but the relation "more substantial SMB changes when closer to the margin of the ice" does not strictly hold. Thus, "most" was changed to "considerably" (l. 230).

L208: Please rephrase:"...variables on the SMB across the entire GrIS and three regions

previously used by Zolles and Born (2019)(Fig. 7).

We have adopted the phrasing, except for "and show three regions".

L219-222: You may rephrase and split the sentence. I suggest:"... simulation (Fig. 7) driven by the SSP585 scenario. It highlights non-linearities that amplify the SMI reduction. For example, air temperature and precipitation often covary so that the increased precipitation compensates the increased melt only partly. If heavier precipitation delivers more rain, the energy required to refreeze the additional rain in the snowpack increases its heat."

Done

L222-224: Here, you create the impression that the combination of a transient air temperature and longwave radiation explains the non-linearities. I believe you, but you have performed only simulations where one variable is transient and not two, or? If this is the case, please formulate it more carefully. You may add at the beginning of the sentence: We conclude that when the air temperature and longwave ….".

Done

L223: Please be very clear about the distinction between observation deduced from measurements and simulated results. Hence, replace "is observed" with "is detected."

Done

L225: Please clarify if you mean: "on the vertical temperature gradient in the snow."

Thank you for pointing out that this is unclear. We have clarified it, whereas we wanted to refer to the temperature difference between air and snow surface (l. 266).

L226-228: Do you mean: "Since the sublimation is driven by the saturation pressure difference between lower atmosphere and surface, sublimation increased for a higher dew point temperature while it is reduced for a warmer surface."

We have adopted most of this wording (l. 266-267).

L226-228: Unclear sentence. Please improve.

We have clarified the last sentence of the paragraph (l. 268-270).

L243-244: You may shorten: "As a consequence, the scenario uncertainty is reduced (Fig. 8b)."

Done

L274: Since you talk about the unknown future, I would like to suggest a slight change:" temperature changes are probably exacerbated."

Done

L301-302: A sentence is missing stating that increased blocking leads to amplified mass loss. Would you please add an appropriate citation?

Done (l. 352)

L303: I suggest rephrasing: "Therefore, our future SMB projections are conservative

because the climate models do not fully represent the expected increase of the Greenland block in a warming climate."

Done

*Appendix*

L327: I suggest rephrasing: "show strong oversaturation of humidity in areas with very low temperatures while only small oversaturation occurs in nature... ."

Done

L342: Please add "t":" of these uncertainties contributors and time *t* as indicated … ."

Done

*Figures*

The figures are in general of high quality and well prepare. However, some technical issues remain, which are discussed below.

Some figures having a time axis do not show the evolution until 2100 (Figure 5, 8). Is this an artifact? If so, please correct it.

The reason is that the variance splitting approach is applied to the decadal running means of the yearly SMB, which is not valid until 2100. We have added this information to the captions of Fig. 5 and 8.

Figure 1: Please state the meaning of the vertical line in all subplots. Would it be possible to indicate the used EraInterim data period in the plots?

We have stated the meaning in the figure caption: The used ERA-interim data period is left of the vertical line (1979-2014).

Are these line plots the actual medians, or are these the climate model anomalies relative to their related historical or pre-industrial climate states? If so, please clarify.

Thank you for pointing out that this is unclear. They are not anomalies. We have stated in the caption of Fig. 1 that these are interpolated and bias-corrected GCM data.

In the caption, you may state that the radiative fluxes are "at the surface."

Thank you for the comment, but "surface downwelling short/longwave radiation" are the designations in the CMIP6 database. We think that re-formulating this could lead to confusion.

Since you use for the first time the unit "kg m$^{-2}$" for precipitation, you may add the following note: "Please note the precipitation unit, 1 kg m$^{-2}$ equals 1 mm(WE); WE=Water equivalent."

Done

Figure 3: The labels indicating the subplot (a-d) are below the black ground. Please repair.

Would you please move the column labels above the top figure row? The subplot in the lowest row shows only black and grey patterns and does not resemble the values of the

related colorbar. Please fix. In the figure caption, I consider the last sentence using the wording "meaningful" as jargon. Please improve.

We have moved the labels. The grey patterns should show the grid points in which the SMB is close to zero. We have changed the grey patterns to hatching. We have also reformulated the figure caption.

Figure 5: Great figure, but would it be more appropriate to represent the square root of the variance/standard deviation in subplot (a)? In this case, the values could be directly compared to recent mass balance estimates and sea-level potential. If you would like to keep your variance y-axis, you may add a second axis with the square root values.

The variance is shown because the law of total variance allows to add the split variances. This is not true for the standard deviation, so we do not show it in this figure.

Figure 6: The subplot labels are cover by the black subfigure's background. Please fix.

We do not understand this comment because in our file, the subfigures do not have a black background.

As mentioned for Figure 5, would it be possible to present the square root of the variance too?

See above in our comment to Fig. 5.

Why do you not use for the right column the last complete decadal period 2091-2100?

As stated above, the reason is that the variance splitting approach is applied to the decadal running means of the yearly SMB, which is not valid until 2100. We have revised the explanation in the caption of Fig. 6.

Figure 7: In the subplot (e), only the region "East" is shown. Please repair.

We do not understand this comment because in our file, all regions are clearly labeled.

*Table*

Table 1: Please separate the units corrected by replacing "kgm$^{-2}$" with "kg m$^{-2}$", for instance.

Unfortunately, the unit kg m$^{-2}$ is wrong, it should have been Gt yr$^{-1}$. We apologize for this mistake which has now been revised, including the separation of the units.

**Bibliography**

Morlighem, M., Williams, C. N., Rignot, E., An, L., Arndt, J. E., Bamber, J. L., Catania, G., Chauché, N., Dowdeswell, J. A., Dorschel, B., Fenty, I., Hogan, K., Howat, I., Hubbard, A., Jakobsson, M., Jordan, T. M., Kjeldsen, K. K., Millan, R., Mayer, L., Mouginot, J., Noël, B. P. Y., O'Cofaigh, C., Palmer, S., Rysgaard, S., Seroussi, H., Siegert, M. J., Slabon, P., Straneo, F., van den Broeke, M. R., Weinrebe, W., Wood, M. and Zinglersen, K. B.: BedMachine v3: Complete Bed Topography and Ocean Bathymetry Mapping of Greenland From Multibeam Echo Sounding Combined With Mass Conservation, Geophys. Res. Lett., 44(21), 11,051-11,061, doi:10.1002/2017GL074954, 2017.

References

Krebs-Kanzow, U., Gierz, P., Rodehacke, C. B., Xu, S., Yang, H., and Lohmann, G.: The diurnal Energy Balance Model (dEBM): a convenient surface mass balance solution for ice sheets in Earth system modeling, The Cryosphere, 15, 2295-2313, https://doi.org/10.5194/tc-15-2295-2021, 2021.

Krebs-Kanzow, U., Gierz, P., and Lohmann, G.: Brief communication: An ice surface melt scheme including the diurnal cycle of solar radiation, The Cryosphere, 12, 3923-3930, https://doi.org/10.5194/tc-12-3923-2018, 2018.

Schaffer, J., Timmermann, R., Arndt, J. E., Kristensen, S. S., Mayer, C., Morlighem, M., and Steinhage, D.: A global, high-resolution data set of ice sheet topography, cavity geometry, and ocean bathymetry, Earth System Science Data, 8(2), 543-557, https://doi.org/10.5194/essd-8-543-2016 , 2016.

Sodemann, H., Schwierz, C., and Wernli, H.: Interannual variability of Greenland winter precipitation sources: Lagrangian moisture diag-nostic and North Atlantic Oscillation influence, Journal of Geophysical Research: Atmospheres, 113, 2008.

Zolles, T. and Born, A.: Sensitivity of the Greenland surface mass and energy balance to uncertainties in key model parameters, The Cryosphere, 15, 2917-2938, https://doi.org/10.5194/tc-15-2917-2021, 2021.

Zolles, T., Maussion, F., Galos, S. P., Gurgiser, W., and Nicholson, L.: Robust uncertainty assessment of the spatio-temporal transferability of glacier mass and energy balance models, The Cryosphere, 13, 469-489, https://doi.org/10.5194/tc-13-469-2019, 2019.

**Response to Anonymous Referee #2**

We would like to thank the Anonymous Referee #2 for the constructive review.

General Comments
The manuscript is well written and well laid-out. I find the authors' approach to be overall logical. I believe the manuscript is well suited for the cryosphere and should be accepted. However, I have some general and specific points that I think the authors should address before the manuscript can be published:

(1) I think the authors should provide some additional detail about the parameterizations used in BESSI and whether the parameter changes used in the BESSI sensitivity studies really representative of the uncertainty in modeling the Greenland snow and ice surface. I would imagine that comparing multiple SMB models across simulations would add to uncertainty in this component. There are processes (e.g. the evolution of bare ice albedo) that are still not well understood or included in models, and which could add uncertainty in future projections. The authors should discuss these potential caveats in further detail especially with regard to their conclusion that the uncertainty associated with the snow model parameters is small.

We agree that in contrast to the sampling of current climate models, our assessment of potential biases in SMB models is incomplete. The primary goal of this study is to quantify the uncertainty arising from different plausible climate simulations, but we agree that this caveat should be discussed. We have added a short description of the parametrisations in Sect. 2.1 (l. 76-82). In the discussion (l. 345-347), we have added the reference to Ryan et al. (2019) who state that the snowline and associated albedo is not represented well in current models. In Sect. 3.2, we mention the performance of BESSI compared to other SMB models studied by Fettweis et al. (2020), and the spatial resolution as possible contributions to the snow model uncertainty (l. 209-213).

(2) The authors utilize a single ensemble member from each GCM to evaluate the inter-model uncertainty. The inter-model uncertainty is therefore influenced to some extent by the internal variability of each model. The question this raises is whether the inter-model variability is larger than the internal variability of any one model. The authors' assumption seems to be that the spread of the single ensemble members from each model is indicative of the uncertainty caused by the inter-model uncertainty. It is not clear whether this is the case. For a more definitive result, it would be useful to perform some additional experiments: (1) Forcing BESSI with the ensemble mean from each simulation rather than single ensemble members, and (2) Forcing BESSI with multiple ensemble members from a single model to compare the inter-model vs. single-model spread. It would be useful to have at least one of these additional simulations if the authors think this makes sense.

We would like to thank the Referee for the valuable hint that selecting only one ensemble member can result in errors in assigning variance to inter-model uncertainty or internal variability. The suggested alternative (1) would lead to a non-negligible bias in SMB, because the averaging reduces the variability (Zolles and Born, in prep.).
Thus, we have conducted the alternative (2): We have forced BESSI with 10 ensemble members of the GCM ACCESS-ESM1-5 that have different initial conditions, for the four

selected climate scenarios and 16 BESSI parameter combinations. We have chosen this particular GCM because its simulated SMB is close to the GCM median, and several ensemble members are available. Only 10 of the ACCESS-ESM1-5 realisations have all of BESSI's input variables available for our four selected climate scenarios. We have applied the method by Hawkins and Sutton (2009), substituting the different climate models for the different realisations, except that we have chosen a 3$^{rd}$ instead of 4$^{th}$ degree polynomial to avoid overfitting. The results (corresponding to Fig. 5 in the manuscript) are shown below:

[Figure]

The total variance of the ACCESS-ESM1-5 ensemble and all of its components are smaller than those in the GCM ensemble because the forcing data are subject to less variability. The relative uncertainty attributed to the realisations is up to 35%, and drops to 8% in the end of the century. This quantity is a measure for the amount internal variability wrongly attributed to the GCM uncertainty (Lehner et al., 2020). Therefore, the relative GCM uncertainty and internal variability in Fig. 5 of the manuscript are not reliable in the first decades. However, in the end of the century, we can still state that the GCM uncertainty is greater than the climate scenario uncertainty, because the realisation uncertainty is low.

We have added a written summary these results to the manuscript in Sect. 3.2 (l. 216-223).

(3) The authors note that they linearly interpolate GCM input variables onto the 10 km BESSI grid. This seems somewhat problematic, especially in coastal areas where there is a high degree of spatial variability. Because temperature, for example, is dependent on elevation, downscaling methods are often employed to take this into account (e.g. Noël et al…., Fischer et al., 2014). The simple linear interpolation seems likely to lead to biases in the SMB forcing fields. However, as the authors are looking at differences on a broad scale, it might be less important. The authors should discuss the impact that this might have on the results, and if possible test the impact of a different more sophisticated downscaling technique to evaluate the impact on the results.

We agree that the linear interpolation introduces potentially important biases, mostly because the limited resolution does not allow a steep ablation zone, which leads to unrealistically high temperatures being prescribed onto a part of the ice sheet. However, on the scale of the 10 km grid, the biases caused by the differences in topography are corrected: The differences in topography between the GCM and ERA-Interim are corrected through the bias correction performed on the GCM output. Afterwards, the bias caused by

differences between ETOPO used in BESSI and the ERA-interim topography is corrected for temperature and longwave radiation using a constant moist adiabatic lapse rate.
We have added an explanation of the corrections between the different topographies in Sect. 2.2 (l. 110-113). In Sect. 3.2, we have mentioned that elevation differences at the sub-grid scale impact the uncertainty of snow modeling (l. 209-211).

(4) The authors frequently refer to General Circulation Model/Global Climate Model/ Earth System Models simulations as "climate models". However, Regional Climate Models are also "climate models", and the use of the term "climate models" is sometimes confusing. I suggest replacing the term "climate models" with "GCMs" to make clear that these are global simulations.
We now follow this suggestion in the revised manuscript.

Specific Comments
1. Line 17: Add "currently" after "(GrIS)" for clarity.
Done

2. Lines 27-35: Here, following the first sentence, the authors introduce what is done in this study, but then go back to describing previous evaluations in the next paragraph. BESSI is also mentioned in this paragraph, but then introduced later, in the last paragraph of the introduction. I suggest moving this material in this paragraph and combining it with the last paragraph of the introduction, and making clear the different sources of uncertainty that are being addressed.
We have combined this material with the last paragraph of the introduction and changed the order of some sentences to make the new structure more consistent.

3. Line 37: Define the term PDD here.
Done

4. Lines 41-42: Here the authors should make it clear that RCM simulations are used to dynamically downscale GCM simulations, which often do not have the spatial resolution or detailed physical representation of the ice sheet surface needed to simulate SMB reasonably well.
Done (l. 37-38)

5. Line 43: I suggest changing "evaluating" to "downscaling", after clarifying the reason for using RCMs in the previous sentence.
Done

6. Lines 44-46: Here the sentence is a bit unclear. Suggested correction: "In Fettweis et al. (2008), which utilized RCM simulations to project SMB forced with a subset of CMIP3 simulations, a multiple regression for the SMB changes as a function of temperature and precipitation is performed to calculate the SMB changes for CMIP3 simulations that were not used to force the RCM."
We have adopted the suggestion in essence, but slightly shortened (l. 39-42).

7. Lines 52-56: In addition to adding in material from the second paragraph of the introduction, I would suggest briefly explaining what BESSI is and why it is advantageous in this situation as compared with GCM or RCM simulations.
Done

8. Lines 61-62: Please explain how the layers are adjusted. Are the authors referring to the snowfall amount within the grid cell? Is there a maximum thickness of the snow model? If the simulation were run continuously, areas of net positive SMB the amount of snow represented in the model would continuously increase.
Yes, there is a maximum snowmass threshold. When it is exceeded, the mass is removed and it can be assigned to an ice sheet model, which we have not used in this study.
The adjustment takes place by splitting or merging layers depending on the snowmass in each grid cell.
We have added the latter and a reference to Born et al. (2019) and Zolles and Born (2021) to the manuscript (l. 64-65).

9. Line 62: Suggest changing to "15 snow or ice layers" for clarity.
Thank you for suggesting to make this more clear. We have added "snow and firn" to the manuscript.

10. Lines 68-69: What was the result of this comparison?
We have added some of the results of the intercomparison study (l. 71-74).

11. Line 71: Can the authors briefly describe how these parameterizations work?
Done

12. Line 74: "ERAinterim" should be changed to "ERA-Interim" here and throughout, and a brief description and reference should be added.
Done

13. Lines 83-84: Can the authors briefly explain the Bougamont et al. (2005) albedo routine here or at the start of the paragraph? How is the snow vs. ice albedo treated? This is particularly important because the contrast between bare ice and snow albedo plays an important role in the GrIS energy and mass balance (e.g. Ryan et al., 2019).
Ryan, J. C., Smith, L. C., Van As, D., Cooley, S. W., Cooper, M. G., Pitcher, L. H., & Hubbard, A. (2019). Greenland Ice Sheet surface melt amplified by snowline migration and bare ice exposure. Science Advances, 5(3), eaav3738.
We have added an explanation in the second paragraph of Sect. 2.1, which is why we do not need to mention Bougamont et al. (2005) here any longer. We have referred to Ryan et al. (2019) in the discussion.

14. Lines 89-90: Suggest changing "whereas the dewpoint is calculated…" to "with the exception of the dew point, which is calculated…"
Done

15. Line 100: Add "to perform bias correction" after "ratio of the monthly means" for clarity.

Done

16. Lines 104-105: From Figure 1 it seems that shortwave radiation increases, while longwave radiation decreases, contrary to what is said here. (Also see note below about the axis label on Fig. 1e.) Could the authors be referring to the SSP126 simulation? If so it might make more sense to discuss all the simulations shown on the figure. Please clarify and/or revise.

Thank you for pointing out this mistake: In the version of Fig. 1 the Referee has commented on, the labels of the variables were swapped. We apologize, and we have corrected the figure.

17. Figure 1: The axis labels on Fig. 1e seem to be incorrect. I would expect the downwelling SW radiation to have a positive value, and to be of the same order of magnitude as downwelling LW radiation. Also change "Temperature in 2 m" to "Temperature at 2 m" and "Dewpoint in 2 m" to "Dewpoint at 2 m" in the caption.

See comment (16), we have corrected the figure.

18. Line 105: Suggest changing "the larger the increase" to "the larger the change", as there is a larger change in the higher GHG forcing scenarios regardless of the direction of the change.

Done

19. Lines 105-107: Rather, it seems that precipitation shows the least overlap between scenarios, as the model spread is small relative to the scenario spread in that case. Please clarify or revise.

See comment (16), we have corrected the figure.

20. Line 128: What is meant by the "actual" climate simulation? Suggest changing to read "following the temporal distribution of precipitation in the GCM simulation."

Done

21. Line 133: Note which variables were changed here.

Done

22. Lines 145-147: This is a bit confusing. I suggest revising to note that the effect of the larger change in the input variables is a larger cumulative effect on SMB.

Done (l. 168-169)

23. Line 147: Change "snow model" to "BESSI" for clarity.

Done

24. Line 158: Change "increase of" to "change in".

Done

25. Lines 157-158: As mentioned above, I'm confused about the SW and LW radiation changes shown on Fig. 1. The axis labels for SW radiation seem to be incorrect. An

increase in downward SW radiation would be consistent with the decrease in downward LW radiation that is shown. However, an increase in precipitation might be indicative of increased cloud cover, which would increase downward LW radiation and reduce SW radiation. Please revise the figure and/or the text.

See comment (16), we have corrected the figure.

26. Line 158: The SW radiation having little effect on SMB seems contrary to recent studies that suggest recent changes in atmospheric circulation lead to increased downwelling SW radiation and increased melt (e.g. Tedesco et al., 2016; Hofer et al., 2017). However, it could be that the combination of factors and feedbacks, which are not included in these idealized experiments, may play an important role in that case. This could be mentioned in section 3.3. Perhaps it should also be clarified here that SW radiation alone does not influence SMB in the idealized experiments performed.

We have removed the sentence about shortwave radiation at this point of the manuscript, because it is discussed in Sect. 3.3, where we have also added the reference to Hofer et al. (2017) (l. 253-257).

27. Line 172: Is this the decadal running mean of SMB for the entire ensemble? Please clarify.

We thank the Referee for pointing out that this is unclear. We have clarified that it is the decadal running mean of each individual simulation (l. 193).

28. Line 206: Is this a seasonally varying mean?

Yes, we have pointed out that the daily mean of the ERA-Interim time period is used (l. 241).

29. Line 241: Could the authors explain this a bit further? Is this because bare ice is exposed at the surface, and do model assumptions about the snow/ice profile play a role in this feedback? I suppose this effect may fall under the model parameter uncertainty experiments.

In all optimal parameter sets, the snow albedo is considerably larger than the ice albedo. Therefore, we do not discuss this effect as a part of parameter uncertainty. We have added to the manuscript that ice is exposed at the surface (l. 283-284).

30. Line 289: Explain the "multiple regression" a bit further.

We have called it "multilinear regression" for clarity, and explained it a bit further (l. 335-337).

31. Figure 9: The plots here are a bit confusing because the shading on (a) shows the maximum range, while (b) shows the mean and 25 th and 75 th percentiles. Would it be possible to also shade the 25-75 range a slightly different color and show the mean on (a)? Or to show the maximum range on (b)?

We have added the 25-75 range in (a).

32. Line 310: Please define ETOPO and provide a reference.

Done

33. Line 325: Can the authors explain the choice of the 50 m threshold? Is it estimated based on previous assessments?

The 50 m threshold is chosen to exclude snow caps because we limit our study to the GrIS. We have stated this in the manuscript.

Technical Corrections
1. Line 22: Change "are met" to "were met".
2. Line 43: Suggest changing "leaving their use to" to "leaving their use limited to".
3. Line 74: ERAinterim should be changed to ERA-Interim
4. Line 94: Change "in the period of 1979-2014" to "over the 1979-2014 period".
5. Lines 109-111: Suggest adding "(1)" before "The main ensemble…", and "(2)" before "The 'single forcing'…" for clarity.
6. Line 161: Add "there" after "melt increases considerably" for clarity.
7. Line 170: Suggest changing to "climate model uncertainty, climate scenario uncertainty, snow model parameter uncertainty, and internal variability."
8. Line 175: Change "forth" to "fourth".
9. Lines 202-203: Suggest changing to read: "The scenario uncertainty has a similar magnitude as the climate model uncertainty only at the margins of the ice sheet and in the area where the total variance is low."
10. Line 213: I believe the correct term is "desublimation" or "deposition".

We have adopted all technical corrections, except for 2., where we changed it to "limiting their use to", and for 7., where the phrasing does not exist any more after revising the paragraph based on a suggestion of Anonymous Referee #1.

---

## Author Response (AR2)

**Response to Review by Editor**

We would like to thank the Editor for the helpful remarks.

- GCM is rather used for General Circulation Model. Global models are now referred as Earth System Model (ESM). Could you change GCM to ESM in your manuscript?

We follow this suggestion throughout the manuscript.

- l 76-82. Could you list, for albedos and heat parameters, the value in the reference simulation and the range tested in the sensitivity experiments ? A table listing these numbers could be provided in Appendix C.

We have added such a table in Appendix C. To avoid duplication, we have removed the sentences specifying the parameter ranges at the end of Sect. 2.1.

- About ETOPO, please refer rather to ETOPO1 (1km) as there are different versions of ETOPO.

We have clarified this, except that ETOPO1 has a resolution of 1 arc-minute.